# Janus-Pro-R1: Advancing Collaborative Visual Comprehension and Generation via Reinforcement Learning

**Kaihang Pan**[1,*], **Yang Wu**[2,*], **Wendong Bu**[1,*], **Kai Shen**[1,‡], **Juncheng Li**[1], **Yingting Wang**[2], **Yunfei Li**[2], **Siliang Tang**[1,†], **Jun Xiao**[1], **Fei Wu**[1], **Hang Zhao**[2], **Yueting Zhuang**[1]

Zhejiang University[1], Ant Group[2]
{kaihangpan, wendongbu, shenkai, junchengli, siliang}@zju.edu.cn

## Abstract

Recent endeavors in Multimodal Large Language Models (MLLMs) aim to unify visual comprehension and generation. However, these two capabilities remain largely independent, as if they are two separate functions encapsulated within the same model. Consequently, visual comprehension does not enhance visual generation, and the reasoning mechanisms of LLMs have not been fully integrated to revolutionize image generation. In this paper, we propose to enable the collaborative co-evolution of visual comprehension and generation, advancing image generation into an iterative introspective process. We introduce a two-stage training approach: supervised fine-tuning teaches the MLLM with the foundational ability to generate genuine CoT for visual generation, while reinforcement learning activates its full potential via an exploration-exploitation trade-off. Ultimately, we unlock the Aha moment in visual generation, advancing MLLMs from text-to-image tasks to unified image generation. Extensive experiments demonstrate that our model not only excels in text-to-image generation and image editing, but also functions as a superior image semantic evaluator with enhanced visual comprehension capabilities. Project Page: https://janus-pro-r1.github.io.

## 1 Introduction

Recently, multimodal large language models (MLLMs) [5, 12] have emerged to unify comprehension and generation across various modalities within the same next-token prediction paradigm of large language models (LLMs) [33]. Specifically, given a user query about comprehension—"What kind of dog is in this picture [IMG]", or generation—"Generate an image of a cute cat", the model can complete the task by sequentially predicting the appropriate text or image tokens.

However, for current MLLMs, visual comprehension and generation remain largely independent rather than forming a synergistic relationship, integrated into a single model seemingly just for the sake of avoiding parameter redundancy [5, 40, 51]. Especially in visual generation tasks, auto-regression-based methods are intended to unleash the powerful reasoning capabilities of MLLMs to infer more semantic-aligned images in more complex contexts. Unfortunately, even state-of-the-art MLLMs like Janus-Pro [5], still fall short of user expectations for basic text-to-image generation, also limited to accepting only pure text input for generation. This highlights that the robust reasoning mechanisms of LLMs have not been fully integrated to revolutionize, or even advance visual generation. Thus,

---

* Equal Contribution.
‡ Project Leader.
† Corresponding Author.

39th Conference on Neural Information Processing Systems (NeurIPS 2025).

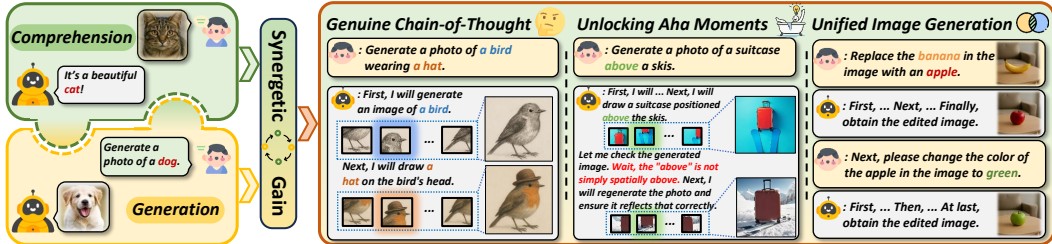

Figure 1: We could bring about three revolutionary benefits for image generation after collaborating the visual comprehension and generation capabilities with MLLMs.

a key question naturally arises: ***Is it feasible to synergize visual comprehension and generation, incorporating the reasoning mechanisms into visual generation?*** Once such collaboration is achieved, the MLLM can seamlessly combine and switch between its comprehension and generation capabilities, bringing about three revolutionary benefits for image generation, as shown in Figure 1.

**1) Genuine Chain-of-Thought (CoT)** [54]: A genuine CoT in MLLMs should be self-driven by the model's deep thinking within a unified next-token prediction framework based on the causal dependency of tokens. The visual comprehension and generation capabilities are naturally linked to form an interleaved text-image reasoning chain under the spontaneous scheduling of the MLLM, which can be treated as a CoT that truly helps produce more refined images.

**2) Unlocking Aha Moments** [16]: Genuine self-driven CoT further endows MLLMs with the ability of self-correction, unlocking the Aha Moments. After generating the initial image, the MLLM leverages its comprehension capability to reflect on the current generation. Once errors are detected, it re-engages its visual generation capabilities to re-produce images that better meet user requirements.

**3) Enabling Unified Image Generation**: The emergence of the above two benefits signifies that the model can effectively collaborate its visual comprehension and generative abilities. This not only enhances its performance in text-to-image tasks, but also enables flexible unified image generation [56] for any complex situational purposes, such as image editing.

Furthermore, the collaboration between visual comprehension and generation should yield **mutual benefits**. This means that as visual comprehension evolves the capabilities of visual generation, it also enhances its own performance in the process.

Motivated by the above goals, we develop Janus-Pro-R1, advancing text-to-image generation into an introspective process and taking a step toward unified image generation with the collaborative co-evolution of visual comprehension and generation capabilities. The visual generation capability anchors the lower bound, ensuring the model can produce appropriate images. Meanwhile, the visual comprehension capability elevates the upper bound, enabling robust reasoning chains that unlock Aha Moments in image generation.

Specifically, our training process comprises two stages. **In the first stage**, we employ supervised fine-tuning (**SFT**) to endow MLLMs with the foundational ability to construct a genuine reasoning chain for visual generation that triggers Aha moments. To achieve this, we break down the CoT generation into several sub-skills with a mixed training approach, upgrading text-to-image generation into an iterative introspection mechanism. After each round of generation, the MLLM self-reflects on whether the generated image meets the requirements and repeats the generation if it does not. The model's spontaneous self-reflection during this process forms genuine CoT for visual generation, triggering Aha Moments where the model redirects its reasoning back onto the correct path.

However, training solely based on SFT tends to naively mimic the training distribution rather than performing true reasoning. Therefore, **in the second stage**, we treat image generation as a long token-level Markov decision process and perform reinforcement learning (**RL**) based on GRPO algorithm [45]. Without any ground-truth images, we encourage the model to spontaneously collaborate its comprehension and generation capabilities for introspective text-to-image generation, also designing a bi-level QA-based reward function for optimization. It represents a trade-off between exploration and exploitation [48]: we avoid explicitly teaching the model how to solve the problem but encourage autonomous exploration, while we still provide it with appropriate incentives to facilitate effective exploitation. And we find that RL enables the MLLM to autonomously develop advanced CoT for image generation, evolving from initial imitation to genuine reasoning.

Thanks to the above training strategy, with Janus-Pro as the backbone, Janus-Pro-R1 achieves stronger text-to-image performance on various T2I benchmarks with the collaboration between the visual comprehension and generation capabilities. More remarkably, the capability synergy endows the MLLM with the potential for unified image generation tasks (*e.g.*, image editing) and further enhances its role as an excellent image semantic evaluator, bolstering its visual comprehension capabilities. Our main contributions are threefold:

- We introduce a two-stage training paradigm, collaborating the visual comprehension and generation capabilities within the MLLM to construct a genuine chain-of-thought and unlock the aha moments for text-to-image generation.

- We further extend text-to-image generation to the scenarios of image editing, unleashing the potential for unified image generation.

- Our model achieves superior performance on both text-to-image generation and image editing tasks, also possessing enhanced image comprehension capabilities.

## 2 Related Work

**Unified Visual Comprehension and Generation.** Recent studies focus on unifying visual comprehension and generation within a unified MLLM. Some efforts [12, 47, 50] cascade external diffusion models after the output of MLLMs for visual generation, while some approaches [5, 39, 64, 52, 35] tokenize images into a discrete space and then conducte unified AR for text and vision. Others [58, 63] try to integrate the objective of AR and diffusion into a single model. However, visual comprehension and generation still remain largely independent without collaborative synergy, and the strong reasoning mechanism of LLMs fails to revolutionize visual generation. In this paper, we aim to promote collaboration between these two capabilities, enabling a genuine CoT for visual generation and unlocking its Aha moments.

**CoT in Visual Generation.** Recently, some studies have attempted to introduce CoT into visual generation. [17] considers the CoT as intermediate images within the DDPM process. [10] proposes a combination of two models: MLLM for reasoning chain generation while the diffusion model interprets the CoT for image generation. Moreover, although [53] and the concurrent work [21] integrate the CoT and image generation within a single model, the CoT is more akin to a forced textual planning of the organization logic for the target image. This represents a rudimentary form of thinking, failing to genuinely drive the MLLM's deep reasoning and introspection for image generation. We argue that a true CoT should emerge spontaneously from the model's deep thinking, naturally collaborating its visual comprehension and generation capabilities into an interleaved image-text reasoning chain, which could unlock the Aha moment in visual generation.

## 3 Method

In this section, we introduce how to collaborate visual comprehension and generation abilities to achieve introspective image generation with genuine CoT that unlocks the Aha moments. We outline two training stages (Figure 2): supervised fine-tuning (§ 3.1) and reinforcement learning (§ 3.2). Finally, we further endow MLLMs with advanced capabilities of unified image generation (§ 3.3).

### 3.1 Supervised Fine-Tuning

Similar to teaching complex tasks in everyday scenarios, during the SFT phase, we break down the Chain of visual generation reasoning into several subtasks with a mixed training approach. By doing so, the MLLM acquires a preliminary ability to continuously imitate specific subtasks based on the current context, finally exploring a coherent reasoning chain for visual generation.

**Data Preparation.** We aim to construct an image-text collection $\mathcal{C} = \{\mathcal{P}_i : \{(\mathcal{I}_i^j, \mathcal{S}_i^j, \mathcal{R}_i^j)\}_{j=1}^M\}_{i=1}^N$ as the supervised training data, where each prompt $\mathcal{P}$ corresponds to $M$ images $\mathcal{I}$. For each image-text pair, there is an associated semantic consistency score $\mathcal{S} \in [0,1]$ and a detailed reason $\mathcal{R}$ for semantic matching ($\mathcal{S} > 0.5$) or non-matching ($\mathcal{S} < 0.5$), where $S^0 > S^1 > ... > S^M$.

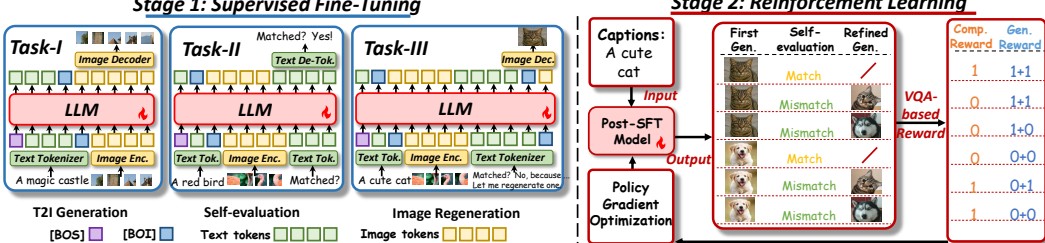

Figure 2: We include two training stages to unlock aha moments with CoT in visual generation: supervised fine-tuning and reinforcement learning.

Specifically, we first leverage the Qwen model [1] to develop a total of $N = 200,000$ prompts. For each prompt, we employ the FLUX [23] & Janus-Pro [5] for text-to-image generation, and then utilize the InternVL2.5-26B model [6] to evaluate whether the generated image and text are semantically consistent. We record the probability to answer "Yes" as $P_{\text{Yes}}$ and "No" as $P_{\text{No}}$, with consistency score $\mathcal{S} = P_{\text{Yes}}/(P_{\text{Yes}} + P_{\text{No}})$. More details are shown in Appendix B.

**Mixed Training.** We first decompose the process of visual generation CoT, which elicits the aha moment, into three steps and designed three basic sub-tasks:

**Task-I: Text-to-Image Generation**: In $\mathcal{C}$, we select text-image pairs with $\mathcal{S} \geq 0.8$ for text-to-image training. The objective function is $p(y^{\mathcal{I}}) = \sum_{j=1}^{S^{\mathcal{I}}} \log P_\theta(y_j|y_{<j}, \mathcal{P})$, where $y^{\mathcal{I}}$ are tokens of a ground-truth image with sequence length as $S^{\mathcal{I}}$.

**Task-II: Self-evaluation of Text-Image Consistency**: We aim to enhance the capability of MLLM to determine whether given images are semantically consistent with the text, also providing the rationale for its judgment. To construct training data, for each prompt we first select $(t-1)$ images $\{\mathcal{I}^j, \mathcal{R}^j\}_{j=1}^{t-1}$ with $\mathcal{S} < 0.5$ as the preceding context ($t \geq 1$). Subsequently, we randomly select one positive image ($\mathcal{S} \geq 0.7$) and one negative image ($\mathcal{S} < 0.5$), respectively, as the target image $\mathcal{I}_t$ to be evaluated. The objective function is $p(y^{\mathcal{T}^t}) = \sum_{j=1}^{S^{\mathcal{T}^t}} \log P_\theta(y_j^{\mathcal{T}^t}|y_{<j}^{\mathcal{T}^t}, \mathcal{P}, \{\mathcal{I}^k, \text{No}, \mathcal{R}^k\}_{k=1}^{t-1}, \mathcal{I}_t)$, where $\mathcal{T}^t$ is $(\text{Yes}, \mathcal{R}^t)$ for positive images and $(\text{No}, \mathcal{R}^t)$ for negative images.

**Task-III: Image Regeneration**: We aim to enhance the MLLM to correct previous errors and regenerate accurate images. For each prompt, we first select $t$ images $\{\mathcal{I}_j, \mathcal{R}_j\}_{j=1}^{t}$ with $\mathcal{S} < 0.5$ as the preceding context to simulate previous incorrect generations and corresponding self-reflections. Then we randomly select an image with $\mathcal{S} \geq 0.8$ as the ground-truth $t$-th regenerated image. The objective function is $p(y^{\mathcal{I}^{t+1}}) = \sum_{j=1}^{S^{\mathcal{I}}} \log P_\theta(y_j^{\mathcal{I}^{t+1}}|y_{<j}^{\mathcal{I}^{t+1}}, \mathcal{P}, \{\mathcal{I}^k, \text{No}, \mathcal{R}^k\}_{k=1}^{t})$.

Through the above supervised mixed training, the MLLM learns to integrate different subtasks for introspective text-to-image generation, developing a reasoning chain of deep thinking. Ultimately, it acquires the fundamental capability to trigger Aha moments.

## 3.2 Reinforcement Learning

The RL phase aims to effectively balance the exploration–exploitation trade-off to unlock the full potential of the MLLM. While encouraging the model to autonomously explore reasoning pathways, we also provide appropriate incentives for both its process of visual generation and comprehension as the exploitation, advancing it from mere imitation to genuine reasoning.

**Bi-Level QA-based Rewards.** The overall design philosophy of our reward model is to leverage QA-based visual comprehension models (*i.e.*, InternVL2.5-26B), which will return a consistency score $\text{R}^{QA}(\cdot) \in [0, 1]$ for each text-image pair, to assess the accuracy of the generated image and the MLLM's self-evaluation. And we provide incentives for the final output images along with their preceding reasoning chains, incorporating **bi-level reward scores**: visual generation reward $\text{R}^{Gen}$ and visual comprehension reward $\text{R}^{Comp}$. Assume that we permit the MLLM to perform up to a maximum of $T$-round image generations and the MLLM determines that it has correctly generated the image in the $K$-th round (resulting in a total of $K$ images $\{\mathcal{I}^i\}_{i=1}^K, T \geq K$), we have:

**(1) The generation reward** $\mathtt{R}^{Gen} = \sum_{i=1}^{K-1} \mathtt{R}^{QA}(\mathcal{I}^i) + (T - K + 1) \times \mathtt{R}^{QA}(\mathcal{I}^K)$. The $K$-th image is assigned a potentially larger weight because it represents the final output with higher importance.

**(2) The comprehension reward** $\mathtt{R}^{Comp} = \sum_{i=1}^{K} \left( \mathbf{1} - |\mathtt{R}^{QA}(\mathcal{I}^i) - \mathtt{SE}(\mathcal{I}^i)| \right) \times T/K$, where $\mathtt{SE}(\mathcal{I}) = 0$ or $1$ is the self-assessment on whether the generated image is semantically aligned with the text.

**Policy Gradient.** We leverage Group Relative Policy Optimization (GRPO) [45] as the training algorithm, which has been proven to be highly effective for exploring the reasoning capability of the LLMs. The preliminary of GRPO is given in Appendix A. Specifically, given the input prompt $\mathcal{P}$, we prompt the old policy $\pi_{old}$ to first sample a group of $G$ individual initial images $\{\mathcal{I}_i^1\}_{i=1}^{G}$ and obtain the first-round generation reward for each generated image $\{\mathtt{R}_{i,1}^{Gen}\}_{i=1}^{G} = \{\mathtt{R}^{QA}(\mathcal{I}_i^1)\}_{i=1}^{G}$. Subsequently, the MLLM will assess the semantic consistency between $\{\mathcal{I}_i^1\}_{i=1}^{G}$ and $\mathcal{P}$, returning the evaluation results as $\{\mathtt{SE}(\mathcal{I}_i^1)\}_{i=1}^{G}$. Based on the self-evaluation results, we further obtain the first-round comprehension rewards as $\{\mathtt{R}_{i,1}^{Comp}\}_{i=1}^{G} = \{\mathbf{1} - |\mathtt{R}^{QA}(\mathcal{I}_i^1) - \mathtt{SE}(\mathcal{I}_i^1)|\}_{i=1}^{G}$.

If the MLLM self-assesses that any generated image is not semantically aligned, it will initiate the next round for image regeneration and re-evaluation. Otherwise, it will directly output the image generated in the current round. Assuming that we conduct a maximum of $T$ rounds in a group with the $i$-th sample undergoing $K_i$ rounds, the final reward can be calculated as $\mathtt{R}_i = \mathtt{R}_i^{Gen} + \mathtt{R}_i^{Comp} = \left( \sum_{j=1}^{K_i-1} \mathtt{R}_{i,j}^{Gen} + (T - K_i) \times \mathtt{R}_{i,K_i}^{Gen} \right) + \sum_{j=1}^{K_i} \mathtt{R}_{i,j}^{Comp} \times T/K_i$. Then we can obtain the advantages $\{A_i\}$, where each $A$ measures the relative quality of output compared to the average reward:

$$A_i = \frac{\mathtt{R}_i - \mathrm{mean}\left(\{\mathtt{R}_i\}_{i=1}^{G}\right)}{\mathrm{std}\left(\{\mathtt{R}_i\}_{i=1}^{G}\right)} \tag{1}$$

Finally, we upgrade the policy network parameters by the following training loss:

$$\mathcal{J}(\theta) = \mathbb{E}_{\substack{(\mathcal{P},a)\sim\mathcal{D} \\ \{y_i\}_{i=1}^{G}\sim\pi_{\theta_{old}}(\cdot|\mathcal{P})}} \left[ \frac{1}{\sum_{i=1}^{G}|y_i|} \sum_{i=1}^{G} \sum_{j=1}^{|y_i|} \left( \min\left(\rho_{i,j}A_i, \mathrm{clip}\left(\rho_{i,j}, 1-\varepsilon, 1+\varepsilon\right)A_i\right) - \beta D_{\mathrm{KL}}(\pi_\theta||\pi_{\mathrm{ref}}) \right) \right], \tag{2}$$

where $D_{\mathrm{KL}}$ is the KL divergence to maintain training stability, and $\rho_{i,j}$ is the ratio between the probabilities of $\pi_\theta$ and $\pi_{\theta_{old}}$ for outputting the current token:

$$\rho_{i,j} = \frac{\pi_\theta(y_{i,j} \mid \mathcal{P}, y_{i,<j})}{\pi_{\theta_{old}}(y_{i,j} \mid \mathcal{P}, y_{i,<j})} = \begin{cases} \frac{\pi_\theta\left(y_{i,j}^{\mathcal{I}^t}|y_{i,<j}^{\mathcal{I}^t},\mathcal{P},\{\mathcal{I}_i^k,\mathtt{No},\mathcal{R}_i^k\}_{k=1}^{t-1}\right)}{\pi_{\theta_{old}}\left(y_{i,j}^{\mathcal{I}^t}|y_{i,<j}^{\mathcal{I}^t},\mathcal{P},\{\mathcal{I}_i^k,\mathtt{No},\mathcal{R}_i^k\}_{k=1}^{t-1}\right)}, & \text{output the image in } t\text{-th round} \\[2ex] \frac{\pi_\theta(y_{i,j}^{\mathcal{T}^t}|y_{i,<j}^{\mathcal{T}^t},\mathcal{P},\{\mathcal{I}_i^k,\mathtt{No},\mathcal{R}_i^k\}_{k=1}^{t-1},\mathcal{I}_i^t)}{\pi_{\theta_{old}}(y_{i,j}^{\mathcal{T}^t}|y_{i,<j}^{\mathcal{T}^t},\mathcal{P},\{\mathcal{I}_i^k,\mathtt{No},\mathcal{R}_i^k\}_{k=1}^{t-1},\mathcal{I}_i^t)}, & \text{self-check the image in } t\text{-th round} \end{cases} \tag{3}$$

### 3.3 Towards Unified Image Generation: From T2I Generation to Image Editing

Through the above two-stage training, we introduce true CoT into visual generation, evolving the vanilla text-to-image generation into an iterative introspective process. Essentially, it is the result of the collaboration between visual comprehension and generation, which is a necessary condition for advanced image generation tasks, such as image editing. Similar to the process of introspective image generation, image editing also requires an understanding of the instructions on how to modify existing images and subsequently generate a new image. And it additionally necessitates preserving image fidelity, *i.e.*, maintaining the unedited areas in their original state.

Therefore, we only need to further teach existing models for detail preserving to easily achieve image editing capabilities. We also leverage a two-stage training for image editing: SFT and RL. **During the SFT phase**, we fine-tune the MLLM using a small number of high-quality data to learn the basic requirements of the image editing task, with the objective function denoted as: $p(y^{\mathcal{I}^{out}}) = \sum_{j=1}^{S^{\mathcal{I}}} \log P_\theta(y_j|y_{<j}, \mathcal{I}^c, \mathcal{P}^{ins})$, where $\mathcal{I}^c, \mathcal{P}^{ins}, \mathcal{I}^{out}$ are the input image, the editing instruction, and the edited image, respectively.

**During the RL phase**, we only provide the image condition and the editing instructions without any ground-truth images. We still leverage InternVL2.5-26B as the reward model and design two QA-based rewards: (1) Following score $\mathtt{R}^{Flw} \in [0, 1]$ measures whether the model accurately follows

Table 1: Comparison with state-of-the-art models on GenEval, T2I CompBench and DPG-Bench on zero-shot text-to-image generation. The best results are in **bold fonts** with the second best underlined.

| Method | GenEval | | | | | | | T2I-CompBench | | | DPG-Bench |
| | **Overall**↑ | SingObj↑ | TwoObj↑ | Counting↑ | Color↑ | Pos. ↑ | ColorAttr ↑ | Color↑ | Shape↑ | Texture↑ | Avg↑ |
|---|---|---|---|---|---|---|---|---|---|---|---|
| *Diffusion-based Method* | | | | | | | | | | | |
| PixArt-alpha [4] | 0.48 | 0.98 | 0.50 | 0.44 | 0.80 | 0.08 | 0.07 | 68.9 | 55.8 | 70.4 | 71.11 |
| DALL-E 3 [2] | 0.67 | 0.96 | 0.87 | 0.47 | 0.83 | 0.43 | 0.45 | 81.1 | **67.5** | **80.7** | 83.50 |
| SD3 [9] | 0.74 | 0.99 | 0.94 | 0.72 | 0.89 | 0.33 | 0.60 | - | - | - | 84.08 |
| FLUX.1-dev [23] | 0.66 | 0.98 | 0.79 | 0.73 | 0.77 | 0.22 | 0.45 | - | - | - | 83.79 |
| Sana-1.5 [57] | 0.81 | 0.99 | 0.93 | **0.86** | 0.84 | 0.59 | 0.65 | - | - | - | 84.70 |
| Janus-Flow [31] | 0.63 | 0.97 | 0.59 | 0.45 | 0.83 | 0.53 | 0.42 | - | - | - | 80.09 |
| *MLLM-based Method* | | | | | | | | | | | |
| Show-o [58] | 0.68 | 0.98 | 0.80 | 0.66 | 0.84 | 0.31 | 0.50 | 56.0 | 41.0 | 46.0 | 67.48 |
| SEED-X [12] | 0.49 | 0.96 | 0.57 | 0.29 | 0.82 | 0.14 | 0.15 | 65.7 | 49.2 | 60.3 | - |
| Emu3 [52] | 0.54 | 0.98 | 0.71 | 0.34 | 0.81 | 0.17 | 0.21 | 61.1 | 47.3 | 61.8 | 80.60 |
| DDT-LLaMA [39] | 0.66 | 0.99 | 0.64 | 0.56 | 0.87 | 0.39 | 0.48 | 72.8 | 51.4 | 64.2 | 80.90 |
| VARGPTv1.1 [64] | 0.53 | 0.96 | 0.53 | 0.48 | 0.83 | 0.13 | 0.21 | - | - | - | 78.59 |
| Infinity [18] | 0.73 | - | 0.85 | - | - | 0.49 | 0.57 | - | - | - | 83.46 |
| Janus-Pro [5] | 0.80 | 0.99 | 0.89 | 0.59 | 0.90 | 0.79 | 0.66 | 63.6 | 35.3 | 49.4 | 84.17 |
| GPT-4o [34] | 0.85 | 0.99 | 0.92 | 0.85 | 0.91 | 0.75 | 0.66 | - | - | - | - |
| *MLLM-based Method + CoT* | | | | | | | | | | | |
| Show-o+PARM [17] | 0.69 | 0.97 | 0.75 | 0.60 | 0.83 | 0.54 | 0.53 | 75.0 | 56.0 | 66.0 | - |
| MINT [53] | 0.73 | 0.98 | 0.82 | 0.66 | 0.79 | 0.55 | 0.56 | - | - | - | - |
| GOT [10] | 0.64 | 0.99 | 0.69 | 0.67 | 0.85 | 0.34 | 0.27 | - | - | - | - |
| T2I-R1 [21] | - | - | - | - | - | - | - | 81.3 | 58.5 | 72.4 | - |
| **Ours** (w/o Aha) | 0.83 | 0.99 | 0.93 | 0.60 | 0.89 | 0.82 | 0.74 | 81.2 | 56.3 | 73.3 | 85.02 |
| **Ours** (with Aha) | **0.86** | 0.99 | 0.94 | 0.66 | **0.92** | **0.87** | **0.78** | **83.4** | 59.4 | 75.2 | **85.57** |

the editing request; (2) Preserving score $R^{psv} \in [0, 1]$ indicates how well the model preserves details that are not intended to be changed. And the final reward is calculated as $R^{edit} = 0.5 * R^{flw} + R^{psv}$.

For policy gradient, we also adopt the GRPO algorithm. Given each pair of image condition and editing instruction, we sample a group of $G$ edited images $\{\mathcal{I}_i^{out}\}_{i=1}^{G}$ from the old policy. For the $i$-th image response, we obtain its reward and follow Eq.(1) to obtain its advantage $A_i$. Finally the objective function is the same as Eq.(2). More details are given in Appendix B.

## 4 Experiments

We employ Janus-Pro-7B as the backbone, developing Janus-Pro-R1 and Janus-Pro-R1-Edit, excelling in text-to-image generation (§ 4.1), image editing (§ 4.2), and image semantic evaluation (§ 4.3). More details are given in Appendix C and E.

### 4.1 Introspective Text-to-Image Generation

**Automated Metric Evaluation.** We first conduct an automated metric evaluation on 3 text-to-image benchmarks: GenEval [14], T2I-CompBench [20], and DPG-Bench [19]. The comparison results against both diffusion-based and MLLM-based methods, as well as methods incorporating CoT into image generation, are presented in Table 1. We have the following observations:

**(1)** In most settings, our model surpasses other diffusion-based and MLLM-based baselines, achieving SOTA performance. For example, for GenEval, the overall performance of Janus-Pro-R1 even outperforms GPT-4o. This highlights that our model facilitates better vision-text alignment via unlocking the aha moment. **(2)** Compared to baselines also proposing to incorporate CoT into text-to-image generation, our method achieves superior performance, e.g., consistently outperforming concurrent work T2I-R1 on T2i-Compbench. This highlights the effectiveness of our approach in activating the true CoT for visual generation. **(3)** Compared to the backbone model Janus-Pro-7B, our method achieves performance improvements of 7.5% on Geneval, 47.0% on T2i-Compbench, and 1.7% on DPG-bench, respectively, which underscores the effectiveness of our approach. **(4)** Without activating the Aha moments and directly outputting the initial generated image, performance on the three benchmarks drops significantly, while it remains higher than that of Janus-Pro-7B, which indicates that the introspection mechanism effectively improves the image quality and the training also enhances the first-round image generation.

**Qualitative Examples.** In Figure 3 and Figure 7, we present qualitative examples of Janus-Pro-R1 to trigger Aha moments within its reasoning chains to generate superior images. The model could leverage its visual comprehension capabilities to accurately identify the issues in its initial-generated images, then unleash the visual generation capabilities to output a more accurate image. Even if

Table 2: Main results on PIE-Bench for image editing task.

| Method | T2I Model | Structure Distance ↓ | Background Preservation | | | | CLIP Similarity | |
|---|---|---|---|---|---|---|---|---|
| | | | PSNR ↑ | LPIPS ↓ | MSE ↓ | SSIM ↑ | Whole ↑ | Edited ↑ |
| InstructPix2Pix [3] | SD1.5 [43] | 107.43 | 16.69 | 271.33 | 392.22 | 68.39 | 23.49 | 22.20 |
| MagicBrush [61] | SD1.5 | 26.81 | 26.85 | 66.67 | 171.11 | 83.37 | 23.89 | 20.84 |
| InstructDiffusion [13] | SD1.5 | 74.21 | 20.88 | 142.35 | 353.45 | 76.70 | 24.06 | 21.57 |
| MGIE [11] | SD1.5 | 67.41 | 21.20 | 142.25 | 295.11 | 77.52 | 24.28 | 21.79 |
| Seed-X-Edit [12] | SD-XL [43] | 61.69 | 18.80 | 173.63 | 209.05 | 74.93 | 25.51 | 22.20 |
| EditAR [32] | LlamaGen [46] | 39.43 | 21.32 | 117.15 | 130.27 | 75.13 | 24.87 | 21.87 |
| Janus-Pro-Edit | Janus-Pro [5] | 49.44 | 20.50 | 131.76 | 185.04 | 73.29 | 24.16 | 21.60 |
| **Janus-Pro-R1-Edit (Ours)** | Janus-Pro-R1 | 35.87 | 22.81 | 114.96 | 123.30 | 76.80 | 24.78 | 22.31 |

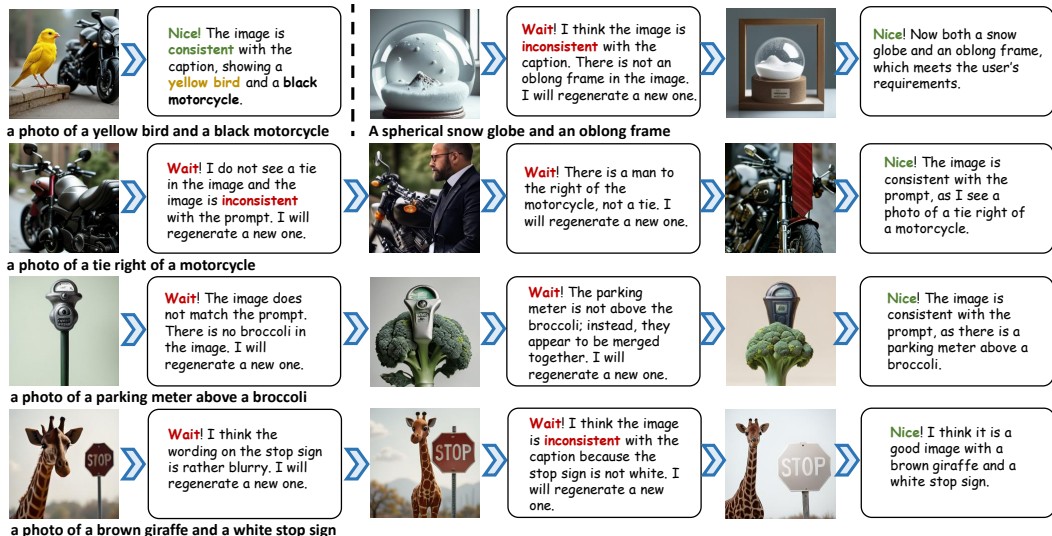

Figure 3: Qualitative examples of introspective text-to-image generation that triggers Aha moments.

the newly generated image still fails to meet the requirements, the model can trigger a second Aha moment, re-evaluating the issues and repeating the visual generation to produce a fully compliant image.

## 4.2 Image Editing

**Automated Metric Evaluation.** We use PIE-Bench [22] as the evaluation benchmark for image editing. We compare with instruction-based large-scale training methods, with structure distance and background preservation metrics reflecting fidelity preservation, while CLIP similarity is used for editability evaluation. As shown in Table 2, most existing methods fail to achieve a good balance between fidelity and editability. While EditAR, which employs the AR framework, demonstrate a relatively better trade-off. Compared to these baselines, Janus-Pro-R1-Edit achieves better overall performance and maintains a good balance between fidelity and editability. Especially compared with EditAR, it excels in most metrics with superior performance.

**Qualitative Examples.** We also conduct a qualitative comparison with both open-source [61, 62] and closed-source [34, 49] leading works on multi-turn editing in Figure 4. Our model supports a wide range of editing operations, with the output images consistently resembling the source images while remaining coherent with the instructions. This stable trade-off between fidelity and editability is rarely achieved in open-source models, and in some cases, our results even outperform closed-source models such as Gemini-2.0 [49].

## 4.3 Janus-Pro-R1 as a Stronger Image Semantic Evaluator

After two-stage training, we find that Janus-Pro-R1 also emerges as an excellent image semantic evaluator. Specifically, it can: (1) Assess whether a given image-text pair is consistent, serving as an evaluation function for text-to-image benchmarking; (2) act as a reward model for text-to-image RL in place of visual comprehension models, enhancing the performance of other text-to-image models.

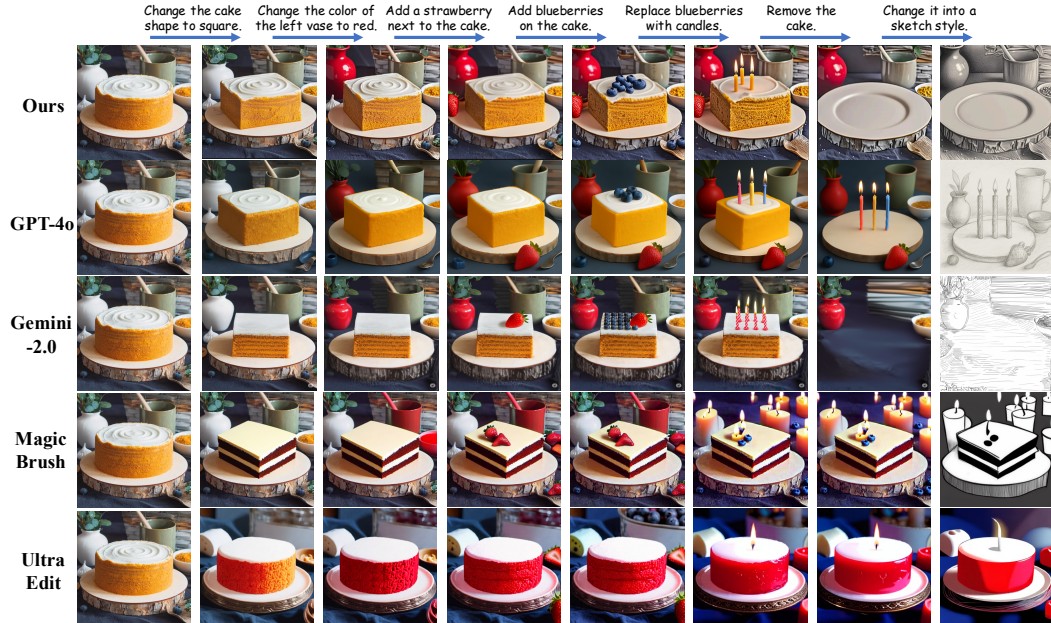

Figure 4: Our model achieve a stable trade-off between fidelity and editability in multi-turn editing.

**Evaluation Metric for Text-to-Image Benchmarking.** We first instruct a text-to-image model Janus-Pro-1B to generate over $2,000$ images for GenEval, and use the original object-focused framework within the benchmark as the standard for text-to-image alignment evaluation. Then we leverage Janus-Pro-7B, Janus-Pro-SFT (7B, only undergoing the first-stage SFT training without RL), and InternVL2.5-8B, and Janus-Pro-R1 (7B) as the evaluation functions, com-

Table 3: The consistency ratio with the standard assessment on GenEval, and the reason reliability score evaluated by GPT-4o, when utilizing different models for text-image alignment evaluation.

| Method | Consistency Ratio on GenEval (%) | Reason Reliability Score by GPT-4o |
|---|---|---|
| Janus-Pro-7B | 72.3 | 76.9 |
| Janus-Pro-SFT (7B) | 72.7 | 78.8 |
| InternVL2.5 (8B) | 79.0 | **92.2** |
| Janus-Pro-R1 (7B) | **81.1** | 91.1 |

paring their results with those of the standard evaluation framework. As shown in Table 3, Janus-Pro-R1 achieves an 81.1% consistency ratio with the standard framework's assessment. In contrast, Janus-Pro-7B, Janus-Pro-SFT-7B, and InternVL2.5-8B consistently exhibit lower consistency ratios compared to Janus-Pro-R1.

Furthermore, we collect several hundred pairs of semantically mismatched text-image pairs. We require the above four models to determine whether the semantics of the image and text are matched and to provide relevant reasons. We then use GPT-4o as the evaluation model to score the reasons provided by the four models from $0$ (unreasonable) to $1$ (reasonable). As shown in Table 3, Janus-Pro-R1 outperforms both Janus-Pro-7B and Janus-Pro-SFT in providing reasons that are deemed more reliable

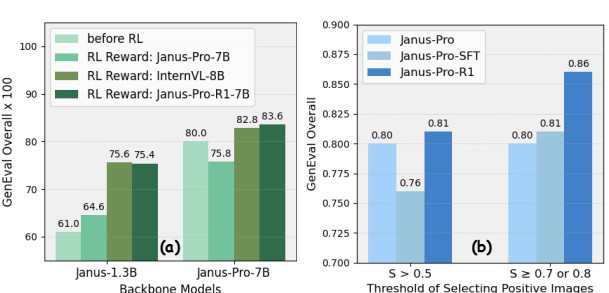

Figure 5: (a) RL Performance with different reward models. (b) Performance with different selecting thresholds in SFT.

by GPT-4o. It even approaches the performance of the comprehension-only MLLM, *i.e.*, InternVL2.5-8B. These results indicate that **(1)** compared to the backbone model and even a comprehension-only MLLM, Janus-Pro-R1 more closely aligns with the standard metrics within GenEval, achieving greater evaluation precision; and **(2)** the visual comprehension capability of our model is significantly enhanced after reinforcement learning. In contrast, SFT only endows the MLLM with the basic capability for combining visual comprehension and generation, resulting in only incremental improvements to its inherent multimodal understanding ability.

Table 4: Zero-shot text-to-image performance on GenEval of different models for in-depth analysis

| | Method | Overall↑ | SingObj↑ | TwoObj↑ | Counting↑ | Color↑ | Pos. ↑ | ColorAttr ↑ |
|---|---|---|---|---|---|---|---|---|
| 1 | Janus-Pro *(7B)* | 0.80 | 0.99 | 0.89 | 0.59 | 0.90 | 0.79 | 0.66 |
| 2 | Janus-Pro-SFT *(7B, w/o aha)* | 0.81 | 0.98 | 0.88 | 0.59 | 0.89 | 0.80 | 0.70 |
| 3 | Janus-Pro-SFT *(7B, with aha)* | 0.81 | 0.99 | 0.87 | 0.57 | 0.90 | 0.81 | 0.72 |
| 4 | Janus-Pro-R1 *(7B, w/o aha)* | 0.83 | 0.99 | 0.93 | 0.60 | 0.89 | 0.82 | 0.74 |
| 5 | Janus-Pro-R1 *(7B, with aha)* | **0.86** | **0.99** | **0.94** | **0.66** | **0.92** | **0.87** | **0.78** |
| 6 | Janus-Pro-SFT1 *(7B, Task-I, w/o aha)* | 0.79 | 0.98 | 0.88 | 0.56 | 0.89 | 0.76 | 0.65 |
| 7 | Janus-Pro-SFT2 *(7B, Task-II+Task-III, with aha)* | 0.76 | 0.98 | 0.86 | 0.55 | 0.85 | 0.73 | 0.61 |
| 8 | Janus-Pro *(1B)* | 0.73 | 0.98 | 0.82 | 0.51 | 0.89 | 0.65 | 0.56 |
| 9 | Janus-Pro-R1 *(1B, w/o aha)* | 0.70 | 0.95 | 0.78 | 0.53 | 0.82 | 0.60 | 0.52 |
| 10 | Janus-Pro-R1 *(1B, with aha)* | 0.71 | 0.98 | 0.80 | 0.51 | 0.84 | 0.59 | 0.55 |

**Reward Model for Text-to-Image RL.** With Janus-Pro-7B and Janus-1.3B [55] as backbone models, we employ Janus-Pro-7B, InternVL2.5-8B [6] and Janus-Pro-R1 as reward models to provide text-image consistency scores as the incentives for text-to-image RL. As shown in Figure 5(a), using Janus-Pro-R1 as the reward model significantly enhances the overall performance of both backbone models on GenEval following reinforcement learning, outperforming the results with InternVL2.5-8B as the reward model. The results highlight the superior visual comprehension capability of our model, which can more accurately assess the text-image semantic consistency.

## 4.4 In-Depth Analysis

**Introspective Text-to-Image Enables Better Image Editing.** To further demonstrate that introspective text-to-image generation with the collaboration between visual comprehension and generation, could enable better image editing, we directly apply the same training paradigm described in § 3.3 to Janus-Pro-7B (we name it as Janus-Pro-Edit). As shown in Table 2, Janus-Pro-R1-Edit consistently outperformed Janus-Pro-Edit on Pie-bench in both instruction following and unedited area preservation, which further validates the effectiveness of our approach.

**Effect of Model Scale to Unlock Aha Moments.** We further employ Janus-Pro-1B as the backbone to explore whether a smaller model could also achieve a breakthrough in visual generation going through the same training paradigm. As shown in Table 4 **Rows 8-10**, in contrast to the significant performance improvement observed in Janus-Pro-R1-7B, Janus-Pro-R1-1B does not achieve superior image generation performance compared to the backbone. And the attempt to activate the Aha moments in this 1B model even compromises its initial image generation capabilities. This suggests that the CoT with deep thinking for visual generation requires a larger parameter size to be effectively handled, which is also an illustration of the scaling law.

**Effect of Data Quality in SFT.** During SFT, when selecting positive images for subtask training, we establish a higher threshold larger than 0.5 of the semantic consistent score ($\mathcal{S} \geq 0.7$ for comprehension tasks and $\mathcal{S} \geq 0.8$ for generation tasks). Though a higher threshold reduces the quantity of available training data, we find that it is crucial for improving performance. In contrast, we also set the threshold as $\mathcal{S} > 0.5$ for selecting positive images in all three subtasks. As shown in Figure 5(b), the decrease in data quality leads to degraded T2I performance of both post-SFT and post-RL models. This highlights that the quality of SFT data is more important than its quantity.

**Synergy of Sub-Tasks in SFT.** We further demonstrate that in SFT, the text-to-image generation capability from Task-I and the regeneration abilities derived from Task-II and Task-III are synergistic. We train two ablation models: Janus-Pro-SFT-1, which undergoes only text-to-image SFT with task-I, and Janus-Pro-SFT-2, which undergoes the mixed SFT of task-II and task-III. As shown in Table 4 **Rows 6-7**, in GenEval, the initial image generation quality of Janus-Pro-SFT is superior to that of Janus-Pro-SFT-1, and the final image generation quality after self-correction is superior to that of Janus-Pro-SFT-2. This indicates that the capabilities of simple text-to-image generation and multi-round image regeneration mutually enhance each other in a synergistic manner.

**SFT Memorizes, RL Generalizes.** We compare the text-to-image performance change after the SFT phase (*i.e.,* Janus-Pro-SFT) and the RL phase (*i.e.,* Janus-Pro-R1). First, as shown in Table 4 **Rows 1-5**, After SFT, the performance improvement compared to the backbone model Janus-Pro-7B is minimal, with image regeneration not significantly improving the quality of the first generated image. This is because SFT tends to enable the model to imitate and memorize some sub-skills to develop the visual generation CoT, lacking the ability of generalization. In contrast, RL enhances

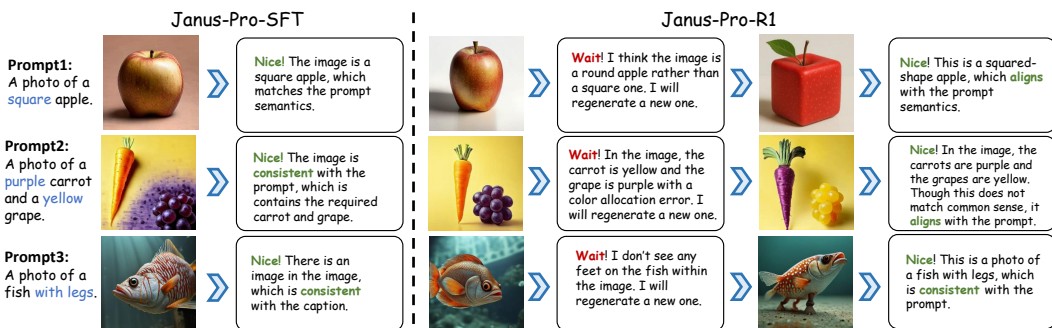

Figure 6: Performance of Janus-Pro-SFT and Janus-Pro-R1 for counterfactual generation.

generalization, significantly improving both the quality of the generated images and ensuring that the introspection process is genuinely effective. Of course, SFT is still essential, as it serves the role as cold-start and provides the foundation for the MLLM to explore reliable visual generation CoTs.

In Figure 6, we further present cases of counterfactual generation to highlight the differences between Janus-Pro-SFT and Janus-Pro-R1. When given counterfactual prompts such as "a square apple", both models initially generate images that do not align with the prompt due to ingrained common-sense. However, Janus-Pro-SFT deems the initial image reasonable as the final output. In contrast, Janus-Pro-R1 identifies the semantic mismatches in the initial image and regenerates a new one that meets the requirements. This demonstrates that RL could generalize the original imitative behavior after SFT to genuine reasoning, thereby better avoiding the generation of hallucinations.

**RL Paves the Way for Genuine Unified Visual Comprehension and Generation.** The essence of unification for comprehension and generation should be reflected in the synergistic enhancement between the two capabilities. So what kind of training paradigm can promote such unification? From Tables 4 (**Rows 1-5**) and 3, we observe that although we can endow models with the foundational ability to collaborate on visual comprehension and generation through SFT, the model appears to merely mechanically mimic and combine these two skills. The combination does not bring about any substantial improvement to either capability. Specifically, the performance of Janus-Pro-SFT-7B in both the visual comprehension task of image-text semantic consistency judgment and the text-to-image generation task shows very incremental improvement compared to Janus-Pro-7B. However, through RL, we encourage the model to spontaneously integrate the two capabilities to explore reasoning pathways, and merely provide incentives as appropriate guidance. We find that the model learns how to better coordinate these two capabilities, resulting in stronger text-to-image generation and image semantic understanding abilities.

Therefore, we argue that *RL holds the potential to unlock genuine unified visual comprehension and generation*. However, the current experiments are insufficient to conclusively validate this assertion. Owing to limited computational resources, our conclusion is based solely on simple image generation tasks and text-image semantic alignment evaluation tasks. Nevertheless, the experiments do provide some preliminary evidence for this conclusion. Given adequate computational resources, we think that large-scale RL could feasibly be employed to achieve a synergistic enhancement of visual comprehension and generation for genuine unification, empowering MLLMs to demonstrate powerful image understanding and generation capabilities comparable to GPT-4o.

## 5 Conclusion

In this paper, we enable MLLMs to form a genuine CoT via deep thinking, advancing text-to-image generation into an iterative introspective process, thereby unlocking the Aha moments in visual generation. We introduce a two-stage training paradigm: SFT teaches the MLLM with the foundational ability to generate visual generation CoT with task decomposition, while RL effectively balances the exploration–exploitation trade-off to unlock its full potential. On this basis, we further endow the MLLM with unified visual generation capabilities, such as image editing. Extensive experiments demonstrate that our model achieves superior performance in both text-to-image generation and image editing tasks, also serving as an excellent image semantic evaluator.

## Acknowledgements

This work was supported by the Key R&D Projects in Zhejiang Province (No. 2024C01106, 2025C01030), the NSFC (62272411), the Zhejiang NSF (LRG25F020001) and Ant Group, My Bank.

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

## Appendix Overview

The anonymous project of our paper is in `https://janus-pro-r1.github.io`. In this supplementary material, we present:

- Preliminary on GRPO in Section A.
- Implementation Details in Section B.
- Evaluation Details in Section C.
- More Experimental Results in Section D.
- Limitation, Future Work, Broader Impacts and Safeguards in Section E.

## A  Preliminary on GRPO

Recently, reinforcement learning [16, 27] has emerged as the primary method for unlocking the reasoning capabilities of LLMs. The study by [45] introduces the Group Relative Policy Optimization (GRPO) framework. GRPO enhances Proximal Policy Optimization (PPO) [44] by eliminating the value function and estimating the advantages in a group-relative manner. Specifically, given the input instruction $q$, the old policy $\pi_{\theta_{old}}$ first samples a group of $G$ individual responses as the response group $\mathcal{G} = \{o_i^1\}_{i=1}^G$. We input each response with the group into the reward function to obtain the individual reward $\texttt{R}_i$. We then calculate the advantages $\{A_i\}_{i=1}^G$, where each $A_i$ measures the relative quality of output compared to the average reward:

$$A_i = \frac{\texttt{R}_i - \text{mean}\big(\{\texttt{R}_i\}_{i=1}^G\big)}{\text{std}\big(\{\texttt{R}_i\}_{i=1}^G\big)} \tag{4}$$

The GRPO method employs a clipped objective function, similar to PPO, and introduces a KL divergence constraint that compares the current policy $\pi_\theta$ with the reference model $\pi_{\theta_{ref}}$ into the loss function, as follows:

$$\mathcal{J}(\theta) = \mathbb{E}_{\substack{(q,a)\sim\mathcal{D} \\ \{o_i\}_{i=1}^G \sim \pi_{\theta_{old}}(\cdot|q)}} \left[ \frac{1}{\sum_{i=1}^G |o_i|} \sum_{i=1}^G \sum_{j=1}^{|o_i|} \left( \min\Big(\rho_{i,j}A_i, \text{clip}\Big(\rho_{i,j}, 1-\varepsilon, 1+\varepsilon\Big)A_i\Big) - \beta D_{\text{KL}}(\pi_\theta||\pi_{\theta_{ref}}) \right) \right], \tag{5}$$

where $D_{\text{KL}}(\pi_\theta||\pi_{\theta_{ref}}) = \frac{\pi_{ref}}{\pi} - \log\frac{\pi_{ref}}{\pi} - 1$ is the the KL divergence to maintain training stability. And $\rho_{i,j} = \frac{\pi_\theta(o_{i,j}|q,o_{i,<j})}{\pi_{\theta_{old}}(o_{i,j}|q,o_{i,<j})}$ is the ratio between the probabilities of $\pi_\theta$ and $\pi_{\theta_{old}}$ for outputting the current token.

## B  Implementation Details

### B.1  Supervised Fine-Tuning

For SFT, we construct an image-text collection $\mathcal{C} = \{\mathcal{P}_i : \{(\mathcal{I}_i^j, \mathcal{S}_i^j, \mathcal{R}_i^j)\}_{j=1}^M\}_{i=1}^N$ as the supervised training data, where each prompt $\mathcal{P}$ corresponds to $M$ images $\{\mathcal{I}_i^j\}_{j=1}^M$. For each image-text pair, there is an associated semantic consistency score $\mathcal{S} \in [0,1]$ and a detailed reason $\mathcal{R}$ for semantic matching ($\mathcal{S} > 0.5$) or non-matching ($\mathcal{S} < 0.5$), where $S^0 > S^1 > ... > S^M$. Specifically, we first develop a set of seed prompts and leverage the Qwen [1] model to expand them into a total of $N = 200,000$ prompts, covering various types (color, shape, spatial, etc.) with $80,000$ short captions and $120,000$ long captions. For each prompt, we employ the FLUX [23] and Janus-Pro-1B/7B [5] models for text-to-image generation, producing $M = 18$ images per prompt. Subsequently, we utilize the InternVL2.5-26B [6] model as the evaluation model, instructing it with the question of whether the image and text are semantically consistent with the following prompt "Does this image match the description? Please directly respond with yes or no". We record the probability to answer "Yes" as $P_{\text{Yes}}$ and "No" as $P_{\text{No}}$, with $\mathcal{S} = P_{\text{Yes}}/(P_{\text{Yes}} + P_{\text{No}})$. And for each prompt, we sort its associated $M$ images based on their semantic consistency scores in descending order. In addition, to construct the long captions, we first collect some short captions. Subsequently, we prompt Qwen to expand these short captions using the following prompt:

> Please generate the long prompt version of the short one according to the given examples. Long prompt version should consist of 3 to 5 sentences. Long prompt version must sepcify the color, shape, texture or spatial relation of the included objects. DO NOT generate sentences that describe any atmosphere.
> Short: `Case1-Short`.
> Long: `Case1-Long`.
> Short: `Case1-Short`.
> Long: `Case1-Long`.
> Short: `Caption-Shot`.

During supervised fine-tuning, we set $t = 3$ for Task-II and $t = 2$ for Task-III, which means that we allow the MLLMs to generate up to three images (one text-to-image generation and two image regenerations) given a prompt. For mixed training, we set the mixing ratios for Task-I, Task-II, and Task-III as $0.2$, $0.3$, and $0.5$, respectively. More training hyperparameters are detailed in Table5.

Furthermore, given that Janus-Pro integrates two distinct types of image encoders, the understanding encoder is specifically the SIGLIP encoder [60], while the generation encoder is the VQ tokenizer from [46]. For all input images in Task-II, the understanding encoder is employed to encode the input visual embeddings, whereas for all input images in Task-III, the generation encoder is used to encode the input visual embeddings. Additionally, for Task-I and Task-III, each sample's input prompt $\mathcal{P}$ has a 10% probability of being replaced with `[PAD]` tokens.

## B.2 Reward Calculation

The overall design philosophy of our reward model is to leverage QA-based visual comprehension [6, 25, 36, 24, 42, 8, 30, 29] models, which will return a consistency score $\mathtt{R}^{QA}(\cdot) \in [0, 1]$ for each text-image pair, to assess the accuracy of the generated image and the MLLM's self-evaluation. And we provide incentives for the final output images along with their preceding reasoning chains, incorporating **bi-level reward scores**: visual generation reward $\mathtt{R}^{Gen}$ and visual comprehension reward $\mathtt{R}^{Comp}$. Assume that we permit the MLLM to perform up to a maximum of $T$-round image generations and the MLLM determines that it has correctly generated the image in the $K$-th round (resulting in a total of $K$ images $\{\mathcal{I}^i\}_{i=1}^K$, $T \geq K$), we have: **(1) The generation reward** $\mathtt{R}^{Gen} = \sum_{i=1}^{K-1} \mathtt{R}^{QA}(\mathcal{I}^i) + (T - K + 1) \times \mathtt{R}^{QA}(\mathcal{I}^K)$. The $K$-th image is assigned a potentially larger weight because it represents the final output with higher importance. **(2) The comprehension reward** $\mathtt{R}^{Comp} = \sum_{i=1}^K \left(\mathbf{1} - |\mathtt{R}^{QA}(\mathcal{I}^i) - \mathtt{SE}(\mathcal{I}^i)|\right) \times T/K$, where $\mathtt{SE}(\mathcal{I}) = 0$ or $1$ is the self-assessment on whether the generated image is semantically aligned with the text.

To calculate the consistency score, we leverage InternVL2.5-26B [6] as the reward model to provide appropriate incentives, which will return a consistency score $\mathtt{R}_{\mathcal{I}} \in [0, 1]$ for each text-image pair. Specifically, for short prompts, we directly the MLLM with the question "`Does this image match the description? Please directly respond with yes or no`". We record the probability of the model responding with "Yes" as $P_{yes}$ and "No" as $P_{no}$, with the consistency score calculated as $\mathtt{R}^{QA} = P_{yes}/(P_{yes} + P_{no})$.

For long prompts, inspired by [7], we first decompose the prompt into semantic tuples (*e.g.*, attributes and spatial relations) and then generate corresponding yes-or-no questions (*e.g.*, "Is the dog red?"). The reward model is then tasked with performing a VQA task for the prompt and the generated image, returning a score between 0 and 1 for each question in the same manner. The consistency score is obtained by averaging the evaluations of the reward model across multiple questions for a given prompt.

## B.3 Reinforcement Learning

During RL training, we use Janus-Pro-SFT as the backbone model and set the batch size to 128, the group size to 7, and $\beta$ to 0.05. All parameters are tunable. We totally conduct 3000 iterations of post-training optimization. We find that the learning rate is crucial: a learning rate that is too small results in insignificant performance gains, while a learning rate that is too large leads to unstable training. To address this, we design a combined Linear + Cosine learning rate scheduler. The learning rate quickly drops linearly from a peak value to a lower "convert learning rate" at a "convert step",

Table 5: The detailed training hyper-parameters of SFT and RL for both text-to-image and image editing tasks.

| Hyper-parameters | SFT for T2I | RL for T2I | SFT for Image Editing | RL for Image Editing |
|---|---|---|---|---|
| Optimizer | AdamW | AdamW | AdamW | AdamW |
| Optimizer param. | $\beta_1 = 0.9, \beta_2 = 0.95, \epsilon = 1e-6$ | | $\beta_1 = 0.9, \beta_2 = 0.95, \epsilon = 1e-6$ | |
| Peak LR | 2.0e-5 | 6.0e-6 | 2.0e-5 | 1.0e-5 |
| Convert LR | - | 2.0e-6 | - | 2.0e-6 |
| Convert step | - | 400 | - | 300 |
| Min LR | 2.0e-7 | 2.0e-7 | 2.0e-7 | 2.0e-7 |
| LR scheduler | Cosine | Linear+Cosine | Cosine | Linear+Cosine |
| Batch size | 128 | 128 | 128 | 128 |
| Group size | - | 7 | - | 8 |
| $\beta$ | - | 0.05 | - | 0.05 |
| Training Steps | 50K | 3K | 40K | 2.2K |
| Warmup Steps | 1000 | 100 | 1000 | 100 |
| Weight decay | 0.05 | 0.05 | 0.05 | 0.05 |
| Gradient clipping | 1.0 | 1.0 | 1.0 | 1.0 |
| Numerical precision | bfloat16 | bfloat16 | bfloat16 | bfloat16 |
| Resource Usage | 8 NVIDIA A800 | 32 NVIDIA A800 | 8 NVIDIA A800 | 32 NVIDIA A800 |

and then gradually decreases along a cosine curve. However, we still encounter some instability during training, indicated by a downward trend in the reward curve. To address this, we adopt the following measures:

**(1)** When the reward curve dropped sharply, we reduce the learning rate to half or two-thirds of its current value and resume the training;

**(2)** When the reward curve declines gradually, it indicates that the KL divergence constraint imposed by a less capable reference model is limiting further improvement of the current model. To address this, we propose two measures: If the optimization steps applied to the current model relative to the reference model are small, we reduce the KL divergence constraint weight by setting a smaller $\beta$ value and then resume training. Otherwise, we update the reference model to the current model and then resume the training.

The detailed hyperparameters for training are shown in Table 5.

## B.4 Inference

Janus-Pro-R1 can generate both text and images with two separate MLLM heads: an image head for predicting image tokens and a text head for predicting text tokens. We transform the text-to-image generation into an introspective process [41, 37, 38]. During inference, when provided with a prompt, the model initially engages the image head to perform an autoregressive prediction of 576 visual tokens to generate the first image. Subsequently, it switches to the text head to assess whether the generated image aligns with the semantic of the prompt and to predict a relevant reason. If the model determines that the image is consistent with the prompt, the inference is terminated, and the generated image is output. Conversely, if the model deems the image inconsistent, it switches back to the image head to predict another 576 visual tokens for a new image generation attempt. The model then reverts to the text head to reassess the semantic alignment and predict a reason. This cycle of switching between the image and text heads continues iteratively. Of course, to prevent infinite image generation, the model is constrained to generate a maximum of 3 images for each inference.

Furthermore, considering that Janus-Pro incorporates two types of image encoders, during image generation with CoT, we select the image encoder in the following manner: when the model leverages the text head to output text (Task-II), all preceding images are encoded using the understanding encoder. Conversely, when the model leverages the image head to output image tokens (Task-III), all preceding images are encoded using the generation encoder.

Moreover, we set $topk = 50$ for text token sampling and $topk = 4096$ for visual token sampling. Besides, for text-to-image generation (Task-I) and image regeneration (Task-III), during inference we use classifier-free guidance on the logits for autoregressive sampling in a manner similar to [39, 52, 28]. We set the guidance scale to 5.0. It is important to note that during both text-to-image generation and image regeneration, we only mask the input text prompt $\mathcal{P}$ with [PAD] tokens to facilitate classifier-free guidance.

### B.5 Implementation Details of Image Editing

**Supervised Fine-Tuning.**   We first collect image editing training data from [62] and [59]. Given the suboptimal quality of existing image editing datasets, we first conduct a data cleaning process on the training data. With InternVL2.5-26B [6] as the evaluation MLLM, we provide it with before-and-after-editing images along with editing instructions, and then pose the following two questions for each image editing training sample $(\mathcal{I}^c, \mathcal{P}^{ins}, \mathcal{I}^{out})$ inspired by [15]:

**(1) Instruction Following Question:** `Does the edited image follow the instruction? Please directly respond with yes or no.`

**(2) Detail Preserving Question:** `Are the non-edited areas of the edited image consistent with the original image? Please directly respond with yes or no.`

For each question, we recorded the probability of the evaluation MLLM responding with "Yes" as $P_{yes}^{flw}$ or $P_{yes}^{psv}$ and $P_{no}^{flw}$ or $P_{no}^{psv}$, with following score calculated as $S^{flw}(\mathcal{I}^c, \mathcal{P}^{ins}, \mathcal{I}^{out}) = P_{yes}^{flw}/(P_{yes}^{flw} + P_{no}^{flw})$ and preserving score as $S^{psv}(\mathcal{I}^c, \mathcal{P}^{ins}, \mathcal{I}^{out}) = P_{yes}^{psv}/(P_{yes}^{psv} + P_{no}^{psv})$. For each sample, we incorporate it into the training dataset only if both $S^{psv}(\mathcal{I}^c, \mathcal{P}^{ins}, \mathcal{I}^{out}) \geq 0.7$ and $S^{flw}(\mathcal{I}^c, \mathcal{P}^{ins}, \mathcal{I}^{out}) \geq 0.7$. Furthermore, during SFT, each sample's instruction has a 10% probability of being dropped out as `[PAD]` tokens. And we detail the training hyper-parameters for image editing SFT in Table 5.

**Reinforcement Learning.**   During reinforcement learning, we no longer provide ground-truth edited images. Instead, we allow the model to autonomously predict the edited images based on the input images and instructions, only providing proper incentives according to the GRPO algorithm. We still leverage InternVL2.5-26B [6] as the reward model and design two QA-based rewards: (1) Following score $\text{R}^{flw} \in [0, 1]$ measures whether the model accurately follows the editing request; (2) Preserving score $\text{R}^{psv} \in [0, 1]$ indicates how well the model preserves details that are not intended to be changed. The calculation methods for these two scores are identical to those described in the preceding paragraph for computing the following score and the preserving score. And the final reward is calculated as $\text{R}^{edit} = 0.5 * \text{R}^{flw} + \text{R}^{psv}$. , where $\text{R}_{flw}$ is assigned with a relatively low weight as we prioritize enhancing the model's ability to maintain local fidelity. And we find that a smaller weight for $\text{R}_{flw}$ tends to lead the model to output the pre-edited image directly in order to achieve a higher overall reward value. Furthermore, we set the group size to $8$ and the batch size to $128$. The trick that enables stable RL training is similar to that described in Appendix B.3. We provide additional details of the hyperparameters for image editing RL in Table 5.

**Inference.**   During inference, the input images are encoded using the generation encoder. We set $topk = 4096$ and also employ classifier-free guidance on the logits for autoregressive sampling with the guidance scale of $4.0$. We only mask the input text instruction $P^{inc}$ with `[PAD]` tokens to facilitate classifier-free guidance without masking the input image.

## C   Evaluation Details

### C.1   Baseline Methods

For text-to-image generation tasks, we compare Janus-Pro-R1 with diffusion-based and MLLM-based methods, as well as MLLMs incorporating CoT into image generation. The diffusion-based baselines include PixArt-alpha [4], DALL-E3 [2], SD3 [9], FLUX.1-dev [23], Sana-1.5 [57], and Janus-Flow [31]. The MLLM-based baselines include Show-o [58], SEED-X [12], Emu-3 [52], DDT-LLaMA [39], VARGPTv1.1 [64], Infinity [18], Janus-Pro [5], GPT-4o [34]. The MLLM+CoT baselines include Show-o+PARM [17], MINT [53], GOT [10], T2I-R1 [21]. To ensure a fair comparison, for the last category of baselines, we report results without inference-time scaling.

For image editing tasks, we compare Janus-Pro-R1 with instruction-based large-scale training methods, including InstructPix2Pix [3], MagicBrush [61], InstructDiffusion [13], MGIE [11], Seed-X-Edit [12], and EditAR [32].

## C.2 Evaluation Benchmarks

We conduct zero-shot evaluation on 3 existing text-to-image benchmarks: GenEval [14], T2I-CompBench [20], and DPG-Bench [19]. GenEval contains 6 different subtasks of varying difficulty requiring various compositional skills, including `single object` (SingObj), `single object` (TwoObj), `counting`, `colors`, `position`, `color binding` (ColorAttri). And we adopt the metric proposed by [14] for evaluation. Each subtask is scored independently, and the overall score is calculated as the average of all six subtask scores. The T2I-CompBench encompasses three subtasks following [52]: `color`, `shape`, `texture`. Building on prior research, we employ the Blip-VQA score [26] as the evaluation metric. For DPG-Bench, we follow the metrics proposed in [19] to conduct the evaluation.

For the image editing task, we conduct zero-shot evaluation on Pie-bench [22] with 700 examples, covering 10 editing types. Our method uses the source image and editing instructions to predict the target edit. Both reconstruction and text-to-image alignment are evaluated as in [22] with the annotated foreground masks, with structure distance and background preservation metrics reflecting fidelity preservation, while CLIP similarity is used for editability evaluation.

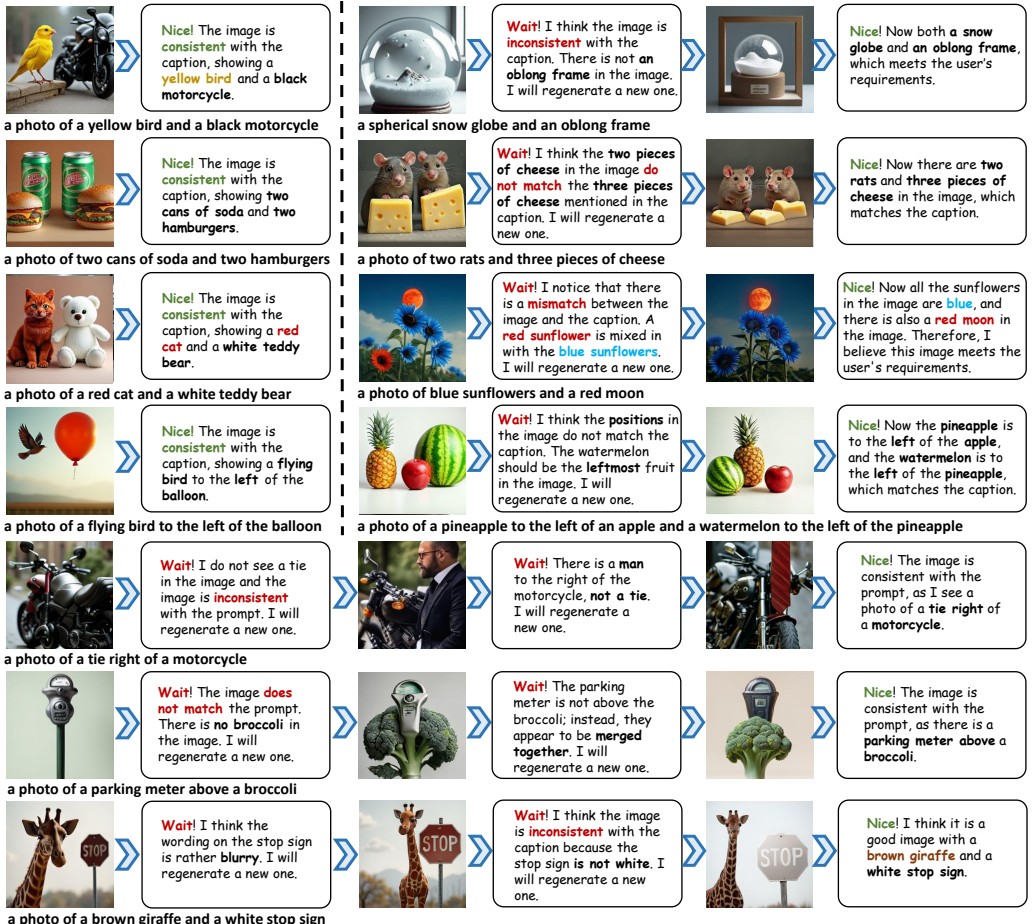

Figure 7: More qualitative examples of introspective text-to-image generation that triggers Aha moments.

# D More Experimental Results

In Figure 7, we present more qualitative examples of Janus-Pro-R1 to trigger Aha moments within its reasoning chains to generate superior images. Our model could leverage its visual comprehension capabilities to accurately identify the issues in its initial-generated images, then unleash the visual generation capabilities to output a more accurate image.

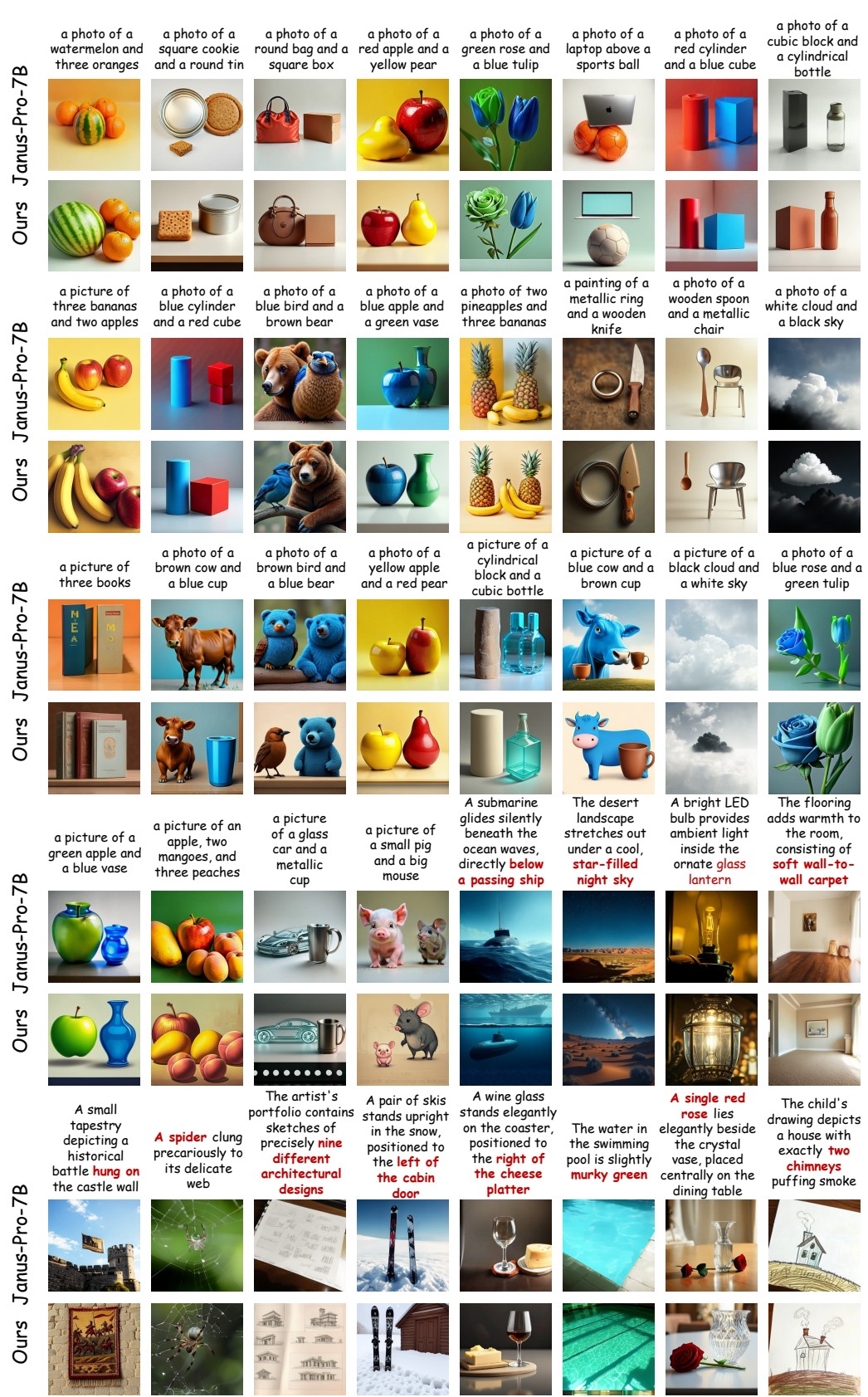

Figure 8: Qualitative Comparisons between Janus-Pro-7B and our Janus-Pro-R1-7B on the final generated image for text-to-image generation task.

Figure 8 presents a direct qualitative comparison between Janus-Pro-7B and our Janus-Pro-R1-7B on the text-to-image generation task, with both short and long captions. It can be observed that compared to Janus-Pro-7B, after unlocking the Aha moments with CoT via a two-stage training paradigm, our model not only generates images that are more semantically aligned with the text but also achieves higher aesthetic quality.

## E    Limitation, Future Work, Broader Impacts and Safeguards

**Limitations.    Firstly,** in the text-to-image generation task, our constructed prompt data contains relatively few instances related to counting. This directly results in our model's counting metric on the Geneval dataset lagging behind SOTA models. To address this issue, we will further specifically construct a set of counting-related data to enhance the corresponding capabilities for counting-related prompts. **Secondly,** in the image editing task, although the existing datasets meet the requirements for image editing, many of the data samples have poor aesthetic appeal. This leads to some edited images lacking in aesthetic quality. To improve this, we plan to use GPT-4o to create a set of high-aesthetic image editing data in the future to enhance the natural appearance of the edited images.

**Future Work.**    In the future, we plan to harness more substantial computational resources to design increasingly complex interleaved text-image generation tasks. We are convinced that reinforcement learning has the potential to bring about revolutionary changes to unified visual comprehension and generation, thereby significantly raising the upper limit of such unification. To this end, we will cover a broader range of visual understanding and visual generation tasks, seamlessly integrate these two capabilities, and tackle the open-ended interleaved text-image generation problem. Ultimately, we hope to transform MLLMs into a more intelligent multimodal dialogue system.

**Broader Impacts.**    This study does not raise any ethical concerns. The research does not involve subjective assessments or the use of private data. Only publicly available datasets are utilized for experimentation.

**Safeguards.**    A major societal concern with this technology lies in its potential for misuse, particularly in fabricating unauthorized images that could lead to misinformation, privacy breaches, and other damaging consequences. To counter these threats, it is crucial to develop strong ethical standards and implement ongoing surveillance.

