# OpenReview forum: "Janus-Pro-R1: Advancing Collaborative Visual Comprehension and Generation via Reinforcement Learning"
_NeurIPS.cc/2025/Conference — NeurIPS 2025 poster_

### Official Review · Reviewer_oSZo · 2025-06-18

**Clarity:** 4
**Significance:** 3
**Originality:** 4
**Rating:** 6
**Confidence:** 4

**Summary:**

The paper proposes to endow Any-to-Any Multimodal LLMs with a capacity to reason before generating images, thereby harmonizing the reasoning capacity in image-to-text understanding with text-to-image generation ability of the same MLLM. Such Chain-of-Thought-based image generation unlocks synergy between image-to-text and text-to-image capacity by allowing iterative feedback and refinement during the image generation process. Starting from a pretrained Any-to-Any MLLM (Janus-Pro), standard recipe of SFT(Supervised Fine-Tuning)-then-RL(Reinforcement Learning) is applied to ellicit CoT-based image generation capacity. For  the SFT stage training, a new training data is collected by generating multiple images given the same prompt and evaluating text-image consistency to support the verificartion and refinement capacity. For RL two reward types are considered: the generation reward for improving the image quality and the comprehension reward for enhacing the verification performance. For editing, the rewards are further divided for update of the edits and preservation of the original features. The final model is shown effective at compositional image generation, image editing and image semantic evaluation tasks.

**Questions:**

1. How would the performance change if one trains image-to-text and text-to-image in separate modules, while initializing the both with the same Janus-Pro weight and train jointly with reinforcement learning? I am curious whether learning to verify leads to better refinement and vice versa. Since this will require extensive ablation study, I am not suggesting to do this in the rebuttal period. Still, can you provide your view on how this intervention will change the outcomes?
2. Is there a step-by-step evaluation result? It would help to see the evaluation results for each generated image_k, k \in [1, … K] in the CoT process.
3. As I suggested above, I think explanation on how this differs from zero-shot CoT-based image generation models [1] or iterative update without verification [2].

[1] https://arxiv.org/pdf/2402.12741
[2] https://arxiv.org/pdf/2410.07171: ICLR 2025

**Ethical Concerns:**

["NO or VERY MINOR ethics concerns only"]

**Final Justification:**

I was generally satisfied with the contribution and presentation of the original draft. My main concern was the lack of an extensive ablation study, which the authors have acknowledged and committed to address with additional experiments. Accordingly, I am raising my score.

**Limitations:**

Yes

**Quality:**

3

**Strengths And Weaknesses:**

Originality: Most previous literature on compositional image generation connect the text-to-image and image-to-text modules or functions in zero-shot manner without a goal to analyze potential synergy between the two. This work not only provides a working implementation of training such unified model, but also analyzes the synergy between each component thoroughly.

Significance: I think synergy between image-to-text and text-to-image is essential in supporting necessity of training the two tasks in a single model. The reasoning-based image generation setup presented in this paper provides a compelling example on the direction: the verification (text-to-image) and refinement (image-to-text) are effectively interleaved for better quality in the downstream task.

Clarity: I found the presentation clear and easy to follow. It would further help gauging the contribution of this work within the current scientific progress to compare the difference with previous compositional and iterative image generation methods.

---

> ### Author Rebuttal · Authors · 2025-07-30
>
> We sincerely thank you for your insightful feedback. We are encouraged by your strong endorsement of our work in terms of Originality, Significance, and Clarity. We will explain your questions point-by-point.
>
> > **(I) Question1**: How would the performance change if one trains image-to-text and text-to-image in separate modules?
>
> **A1:** Thank you for the insightful question! We agree that this is an interesting direction worth further exploration. We will present our view on how this intervention will alter the outcomes.
>
> **(1.1) Disrupting CoT without genuine deep thinking**
>
> Our method integrates visual comprehension (verification) and visual generation (refinement) within a single reasoning trajectory, allowing the model to perform iterative refinement through self-reflection in a flexible and self-driven manner.
>
> When verification and refinement are separated into different models, **the coherent reasoning chain is disrupted. It may fail to genuinely drive the MLLM’s deep reasoning and introspection for image generation.** For the two Janus-Pro models, only the output of one model is passed as input to the other, without preserving the intermediate hidden states that encode crucial reasoning processes. As a result, the Janus-Pro for text-to-image may not genuinely comprehend the image flaws and revision suggestions provided by the Janus-Pro for image-to-text, so the subsequent refinement process may inadequately address the issues identified during verification.
>
> **(1.2) Lacking synergy and mutual enhancement between verification and refinement**
>
> In our method, visual comprehension (verification) and visual generation (refinement) are optimized together in a unified framework and reinforce each other in a synergistic manner. Improved comprehension helps guide more precise generation, while high-quality generation provides richer visual semantics that support better comprehension. This mutual enhancement enables the model to progressively conduct higher-quality verification and refinement.
>
> **In contrast, a decoupled framework with separate modules lacks this synergy, as each module operates in isolation with limited capacity to benefit from the improvements of the other.** As a result, the overall performance may fall short of that achieved by a unified architecture, even though optimizing the two separate modules demands substantially greater training resources.
>
> &nbsp;
>
> Of course, the above analyses will require extensive ablation studies for empirical validation, which we will rigorously conduct after the rebuttal period, and we expect the results to further reveal both the necessity and benefits of unifying image-to-text and text-to-image within a single model. Furthermore, we also regard the mixture-of-transformer (MoT) design introduced in [1] as a highly promising solution: it interlinks the attention functions of the text-to-image and image-to-text modules, ensuring that the two capabilities remain unified within one model while still preserving independent parameters. We view this as a high-ceiling improvement. However, as noted in [1], it demands much more computational resources for training, so we leave its exploration to future work.
>
> *Reference:*
>
> *[1] Emerging Properties in Unified Multimodal Pretraining.*
>
> &nbsp;
>
> > **(II) Question2**: Is there a step-by-step evaluation result?
>
> **A2:** Thank you for your question. We conduct a step-by-step evaluation on GenEval to demonstrate the image quality in the CoT process. Specifically, we evaluate the generated images $I_1$, $I_2$, and $I_3$, corresponding to the first, second, and third rounds of generation, respectively (the maximum round T=3). We set $I_3=I_2$ ($I_3=I_2=I_1$) if the model performs only two rounds (one round) of image generation in a single inference.
>
> As shown in the following table, **our method exhibits consistent performance improvements as the number of reasoning steps increases in the CoT process.** This indicates that our method effectively leverages self-reflection within the long CoT process to iteratively refine the generated images, enabling the model to correct flaws and enhance visual quality through deeper reasoning.
>
> **Table:** Step-by-step GenEval performance of intermediate images $I_k$ generated during the CoT process.
>
> |Image $I_k$|Overall|SingObj|TwoObj| Counting |Color|Pos| ColorAttr |
> |-|-|-|-|-|-|-|-|
> |$I_1$|0.83|0.99|0.93|0.60|0.89|0.82|0.74|
> |$I_2$|0.85|0.99|0.93|0.62|0.91|0.89|0.75|
> |$I_3$|**0.86**|**0.99**|**0.94**|**0.66**|**0.92**|**0.90**|**0.80**|
>
> &nbsp;
>
> > **(III) Question3**: Explanation on how this differs from zero-shot CoT-based image generation models or iterative update without verification.
>
> **A3:** Thank you for your insightful comment! Though our method has a similar iterative optimization paradigm with the zero-shot CoT-based methods and iterative update methods without verification, there are still key differences that distinguish our research from the following three perspectives:
>
> **(3.1) Architecture Advantages: Lightweight Design and Self-Driven Reasoning with Genuine CoT**
>
> Both zero-shot CoT-based image generation models [1] and iterative update without verification [2] typically require assistance from external models, and are hard to perform iterative self-correction in a self-driven manner.
>
> Specifically, MuLan [1] relies on proprietary MLLMs to provide global guidance for image generation, and uses VLMs to decide whether to redraw after generating each object. IterComp [2] introduces a trainable reward model separate from the diffusion model and performs iterative training of both models, making the system complex and heavy. This forced coupling of two distinct models for iterative image generation, **not only fails to genuinely drive the MLLM's deep reasoning, but also results in a relatively heavy framework with limited flexibility.**
>
> In contrast, our method unified the verification and refinement within a single model, bringing two architecture advantages: **(1)** it allows a **genuine CoT** to emerge spontaneously from the model’s deep thinking, naturally collaborating its visual comprehension and generation capabilities into an interleaved image-text reasoning chain; **(2)** it requires no external models, adopting a **lightweight and elegant design** that offers **greater flexibility** than the two aforementioned approaches.
>
> &nbsp;
>
> **(3.2) Synergistic Gains from Unified Comprehension and Generation**
>
> While both methods ([1] and [2]) aim to improve image quality via iterative refinement, they still treat visual comprehension (image-to-text) and visual generation (text-to-image) as **two disjoint processes**, **overlooking the synergistic gains** that arise when visual comprehension and generation are co-optimized within a unified model. For instance, MuLan [1] focuses almost exclusively on refining the visual generation pipeline, thereby neglecting the critical contribution of visual comprehension to image quality. Similarly, although IterComp [2] updates its reward model during training, the optimization of the reward model remains decoupled from that of the text-to-image generator.
>
> While in our method, **unifying visual comprehension (verification) and visual generation (refinement) within a single model can lead to mutual enhancement**, as discussed in prior work [3, 4]. From the comprehension side, a better understanding of visual semantics, such as object identity and spatial layout, helps guide more faithful and coherent image generation. From the generation side, generating visual tokens step by step forces the model to learn detailed visual structures, which in turn improves its ability to comprehend images. As demonstrated in Sections 4.1 and 4.3, **our approach delivers synergistic gains that render Janus-Pro a superior image generator and image-semantic verificator.** Additionally, the visual comprehension and generation capabilities are naturally linked to form an interleaved text-image reasoning chain under the spontaneous scheduling of the MLLM, which can be treated as a CoT that truly helps produce more refined images.
>
> In contrast to our unified method, both zero-shot CoT-based methods and iterative update methods without verification treat comprehension and generation as disjoint processes. As a result, **they are unable to benefit from the synergy to enhance generation quality, ultimately constraining their capacity, especially in complex scenarios.**
>
> &nbsp;
>
> **(3.3) Superior Performance in Image Generation Task**
>
> To better understand the practical impact of these methods, we evaluate two baselines, MuLan [1] and IterComp [2], on the GenEval benchmark. As shown in the following table, **Janus-Pro-R1 significantly outperforms both methods**. Specifically, it surpasses MuLan by 0.34 and outperforms IterComp by 0.18 in overall score. This indicates that our proposed method achieves better image generation performance than zero-shot CoT-based methods and iterative update methods without verification.
>
> **Table:** Performance comparison on the GenEval benchmark.
>
> |Method|Overall|SingObj|TwoObj|Counting|Color|Pos|ColorAttr|
> |-|-|-|-|-|-|-|-|
> |MuLan-SD v1.4|0.38|0.96|0.41|0.15|0.28|0.34|0.14|
> |MuLan-SDXL|0.52 |0.98|0.51|0.17|0.72|0.45|0.30|
> |IterComp|0.68|0.97|0.85|0.63|0.86|0.33|0.41|
> |Janus-Pro-R1 (Ours)|**0.86**|**0.99**|**0.94**|**0.66**|**0.92**|**0.87**|**0.78**|
>
> &nbsp;
>
> *Reference:*
>
> *[1] MuLan: Multimodal-LLM Agent for Progressive and Interactive Multi-Object Diffusion*
>
> *[2] IterComp: Iterative Composition-Aware Feedback Learning from Model Gallery for Text-to-Image Generation*
>
> *[3] SEED-X: Multimodal Models with Unified Multi-granularity Comprehension and Generation*
>
> *[4] Show-o: One Single Transformer to Unify Multimodal Understanding and Generation*
>
> &nbsp;
>
> Once again, we are grateful for your valuable suggestions, which have significantly contributed to improving our paper!

---

> > ### Comment · Reviewer_oSZo · 2025-08-04
> > **Thanks for the Rebuttal**
> >
> > Thank you for your thoughtful response. I believe the additional ablation studies will significantly strengthen the paper’s validity, and I encourage you to include them in the final draft.

---

> > > ### Author Response · Authors · 2025-08-05
> > >
> > > Your valuable suggestions greatly contribute to the quality of our manuscript. Thank you again for your precious time and valuable suggestions!

---

### Official Review · Reviewer_Fzpc · 2025-06-27

**Clarity:** 3
**Significance:** 3
**Originality:** 3
**Rating:** 4
**Confidence:** 4

**Summary:**

The main paper proposes a training framework for text-to-image generation tasks with multimodal large language models (MLLMs) by jointly addressing the visual comprehension and generation.
The proposed method combines supervised fine-tuning with reinforcement learning based on Group Relative Policy Optimization (GRPO), which appears to improve performance on text-to-image generation tasks.

**Questions:**

1. Does the proposed method compromise performance on visual understanding tasks?
As discussed in *MetaMorph [A]*, visual understanding and generation are considered synergistic. Could the proposed approach negatively affect understanding performance?


2. Given that Janus-Pro does not have a particularly large context window, what is the typical value of T used during training?
Additionally, how does the distribution of $K_i$ evolve as training progresses?


**Reference**
[A] S. Tong et al., MetaMorph: Multimodal Understanding and Generation via Instruction Tuning, ICCV 2025.

**Ethical Concerns:**

["NO or VERY MINOR ethics concerns only"]

**Limitations:**

Yes

**Paper Formatting Concerns:**

Nothing

**Quality:**

3

**Strengths And Weaknesses:**

**Strength**
1. The writing is clear.

2. The method is simple and technically sound.


**Weakness**
1. The analysis regarding the judge model appears to be limited. I am curious about the effect of the judge model on the overall performance. Have the authors considered evaluating alternative judge models besides InternVL-26B?

2. The qualitative results focus mostly on relatively simple image editing cases. It is unclear how well the model performs on more challenging tasks, such as multi-object editing.

---

> ### Author Rebuttal · Authors · 2025-07-30
>
> We sincerely thank you for your insightful feedback. We are encouraged that our method is recognized as technically sound. We will explain your concerns point by point.
>
> > **(I) Weakness1:** Why choosing InternVL-26B as the judge model?
>
> **A1:** Thank you for your question.
> Of course, the choice of judge model is crucial to overall performance; we therefore require the judge model to deliver an impartial and objective assessment of image quality. After empirically comparing various vision-language models (VLMs), we found that InternVL2.5-26B is the best open-source VLM for most reliably assessing image–text semantic consistency while achieving the closest alignment with human evaluation.
>
> To illustrate the above conclusion, after collecting hundreds of images generated by Janus-Pro, we score them using human evaluation, and vision-language model (VLM) evaluation, including GPT-4o, InternVL2.5-8B/26B, LLaVA-1.6-34B, ShareGPT4V-13B, and Qwen2.5-VL-32B. We compare the score differences between the VLM evaluation results and the human evaluation results. As shown in the following table, **InternVL2.5-26B has a strong ability to judge fine-grained image-text consistency among open-source models (even nearing the performance of GPT-4o), with its evaluation results very close to those of human evaluation.** Thus, we chose InternVL2.5-26B as the judge model, which can provide an impartial and objective assessment of image–semantic alignment to serve as incentives.
>
> **Table:** Score Difference Between VLM Evaluation and Human Evaluation
> | VLM | Score Difference$\downarrow$ |
> |-|-|
> |GPT-4o| **0.06** |
> |InternVL2.5-26B| ***0.09*** |
> |InternVL2.5-8B | 0.20 |
> |LLaVA-1.6-34B | 0.17 |
> |ShareGPT4V-13B | 0.19 |
> |Qwen2.5-VL-32B | 0.11 |
>
> &nbsp;
>
> We have also experimented with using Qwen2.5-VL-32B as the judge model for RL training, keeping all training settings consistent with the main experiments in our paper. As shown in the following table, switching the judge model from InternVL2.5-26B to Qwen2.5-VL-32B to provide rewards results in a decline in the model’s text-to-image performance. This further demonstrates the soundness of our choice of judge model.
>
> **Table:** Zero-shot text-to-image performance with different judge models
> | Judge Models | GenEval (Overall) | T2ICompbench (Avg) | DPG-Bench |
> |-|-|-|-|
> |Qwen2.5-VL-32B|0.84|70.2|85.02|
> |InternVL2.5-26B|**0.86**|**72.7**|**85.57**|
>
> Of course, we are also considering adopting GPT-4o as the judge model, which could deliver more accurate assessments of generated images. However, the associated API costs exceed our present budget, so we defer this to future work.
>
> &nbsp;
>
> > **(II) Weakness2:** The qualitative results focus mostly on relatively simple image editing cases. It is unclear how well the model performs on more challenging tasks, such as multi-object editing.
>
> **A2:** We apologize that the qualitative results showcase relatively simple image-editing cases in Fig.4; our intention was to highlight that our model can consistently generate images resembling the source images while remaining coherent with the instructions during multi-turn editing. To demonstrate its ability to handle complex tasks, we curated 80 challenging test cases for image editing, including multi-object editing. We conduct a rigorous human evaluation, where human evaluators rate each generated image on an overall 1-to-5 scale, jointly weighing diverse dimensions including fidelity preservation and instruction following.
>
> As shown in the following tables, **our model exhibits superior performance to open-source baseline methods for human evaluation of image editing** (only marginally underperforms GPT-4o), which demonstrates the strong performance of our model. In the next version of our paper, we will provide additional qualitative examples of these more challenging tasks, including multi-object editing.
>
> **Table:** Human evaluation on challenging test cases for image editing
> |Model|Human Score$\uparrow$|
> |-|-|
> |InstructPix2Pix|2.4|
> |MagicBrush|2.1|
> |Ultraedit|2.8|
> |SeedX-Edit|2.6|
> |EditAR|2.5|
> |GPT-4o|**3.8**|
> |**Ours**|***3.2***|
>
> &nbsp;
>
> > **(III) Question1:** Does the proposed method compromise performance on visual understanding tasks?
>
> **A3:** Thank you for the insightful comment!  In fact, our method achieves synergistic gains across both visual generation and understanding tasks, thereby realizing an enhancement in visual understanding capabilities. To address your concern, we will provide detailed experimental results from three perspectives: 1) introducing a fine-grained image understanding experiment, i.e., visual entailment task, to validate the synergistic gain for visual understanding; 2) regarding Janus-Pro-R1 as an image-text semantic alignment evaluator to demonstrate the enhanced visual understanding; 3) Task Janus-Pro-R1 with describing the reasons for semantic misalignment in specific text–image pairs, further validating its enhanced visual understanding capability.
>
> Before presenting the experiments, we first clarify the above evaluation setup. In Meta-Morph, the synergy between visual understanding and generation is concretely demonstrated by showing that mixed training on visual-generation data (e.g., text-to-image) and visual-understanding data (e.g., VQA) jointly improves both tasks. In our work, the continued training of Janus-Pro does not incorporate any dedicated visual-understanding data, such as VQA. Consequently, performance on several VQA benchmarks remains unchanged or even decreases slightly, which aligns with the inherent principles of model training. Our training essentially performs joint optimization of the text-to-image generation task and the visual-understanding task of evaluating image–text consistency. Guided by the rationale underlying Meta-Morph’s conclusion, we therefore select the image–text semantic-consistency task as the basis for our visual-understanding evaluation.
>
> **(3.1) Visual entailment task on SNLI-VE**
>
> To validate that our method enhances visual understanding, we first conduct a zero-shot evaluation on the SNLI-VE benchmark.SNLI-VE is a classical benchmark for the visual entailment task, where each sample consists of an image (as the premise) and a text (as the hypothesis). The model is required to understand the fine-grained semantics in the image and determine whether the text can be entailed by the visual premise.
>
> As shown in the following table, **Janus-Pro-R1 achieves a significant improvement of 20.8 points compared to its backbone model Janus-Pro**. The result indicates that our proposed method also improves visual understanding, achieving a synergistic gain through the unification of understanding and generation.
>
> **Table:** Zero-Shot Accuracy on SNLI-VE benchmark
>
> |Method|Accuracy(\%)$\uparrow$|
> |-|-|
> |Janus-Pro|35.2|
> |Janus-Pro-R1 (Ours)|**56.0**|
>
> &nbsp;
>
> **(3.2) Image-text semantic alignment evaluating**
>
> To further demonstrate the enhanced visual understanding capability of our model, we conduct an additional experiment on the GenEval benchmark. We first instruct a text-to-image model to generate over 2,000 images for GenEval. Then we employ Janus-Pro-R1 and Janus-Pro as the evaluation functions for assessing text-image semantic alignment, comparing their results with those of the standard evaluation framework. More details are given in Section 4.3.
>
> As shown in the following table, Janus-Pro-R1 achieves an 81.1% consistency ratio with the standard framework’s assessment, **exhibiting much higher consistency ratios compared to Janus-Pro**. This indicates that Janus-Pro-R1 more closely aligns with the standard metrics within GenEval, achieving greater evaluation precision and demonstrating stronger visual comprehension ability.
>
> **Table:** Consistency ratio compared with standard assessment in GenEval
>
> |Method|Consistency Ratio(\%)$\uparrow$|
> |-|-|
> |Janus-Pro|73.3|
> |Janus-Pro-R1 (Ours)|**84.2**|
>
> &nbsp;
>
> **(3.3) Image-text misalignment reason description**
>
> Finally, we collect several hundred pairs of semantically mismatched text-image pairs. We require Janus-Pro-R1 and Janus-Pro to determine whether the semantics of the image and text are matched and to provide relevant reasons. We then use GPT-4o as the evaluation model to score the reasons on a 0-to-1 scale (0 = unreasonable, 1 = reasonable).
>
> As shown in the table below, Janus-Pro-R1 surpasses Janus-Pro-7B by producing much more reliable reasons of text-image misalignment, further demonstrating the model’s powerful image–semantic understanding capability.
>
> **Table:** Reason reliability score evaluated by GPT-4o
>
> |Method|Reason Reliability Score$\uparrow$|
> |-|-|
> |Janus-Pro-7B|0.77|
> |Janus-Pro-R1-7B (Ours)|**0.91**|
>
> &nbsp;
>
> > **(IV) Question2:** What is the typical value of T used during training? Additionally, how does the distribution of $K_i$ evolve as training progresses?
>
> **A4:** As Janus-Pro does not have a particularly large context window, we set **T=3**, so the model activates at most two aha moments and thus generates at most three images in a single inference pass.
>
> During RL training, we encourage the model to satisfy the prompt **in as few rounds of image generation as possible**: **the fewer rounds it needs to produce a correct image, the larger the reward we provide.** For instance, if the model can generate a compliant image and perform an accurate self-evaluation in the first round, it could receive the maximum reward score.
>
> Under this reward scheme, **the proportion of cases with K=1 gradually increases, while the proportion of K=2 and K=3 (especially K=3) steadily declines.** Specifically, the distribution shifts from approximately (K=1): (K=2): (K=3) ≈ 70: 18: 12 at the start of training to (K=1): (K=2): (K=3) ≈ 84: 11: 5 by the end of training.
>
> &nbsp;
>
> Thank you once again for your valuable suggestions! We will incorporate the experiments into our revised manuscript.

---

> > ### Comment · Reviewer_Fzpc · 2025-08-03
> > **Thanks for the Rebuttal**
> >
> > Thank you for your clarification. My concerns have been addressed in the rebuttal, and I appreciate the authors’ efforts.

---

> > > ### Author Response · Authors · 2025-08-03
> > >
> > > Thank you again for your precious time and valuable suggestions, which contribute to the quality of our manuscript.

---

### Official Review · Reviewer_xYFR · 2025-07-01

**Clarity:** 2
**Significance:** 2
**Originality:** 2
**Rating:** 4
**Confidence:** 3

**Summary:**

Recent MLLM research aims to unify visual comprehension and generation, but these capabilities remain disjoint, failing to mutually reinforce. This paper proposes collaborative co-evolution of the two, transforming image generation into an iterative introspective process via a two-stage training: supervised fine-tuning for Chain of Thought (CoT) generation and reinforcement learning for exploration-exploitation balance. The approach advances MLLMs from text-to-image tasks to unified image generation. Experimental results demonstrate the model's superior performance in both text-to-image synthesis and image editing scenarios.

**Questions:**

1. The paper uses Qwen to generate 20,000 prompts for constructing the training set. May I ask how the authors ensured the diversity of these prompts?
2. The paper aims to enhance the reasoning ability of Janus-Pro by constructing CoT data. May I ask why it is necessary to use both Flux and Janus-Pro to generate images simultaneously, and what is the significance of image generation by Flux?

**Ethical Concerns:**

["NO or VERY MINOR ethics concerns only"]

**Final Justification:**

The rebuttal has addressed my concerns very well, so I am willing to increase my rating to 4.

**Limitations:**

Yes

**Quality:**

2

**Strengths And Weaknesses:**

**Strengths**
1. It uses a two-stage training paradigm to collaboratively enhance the visual comprehension and generation within MLLMs. By constructing genuine CoT, it unlocks the "aha moments" for T2I generation, enabling a more intelligent and coherent generation process.
2. The method extends the traditional T2I generation to image editing scenarios, thus unleashing the potential for unified image generation. This expansion broadens the practical applications of MLLMs in the visual domain.
3. The model based on this method achieves a new SOTA performance in both T2I generation and image editing, demonstrating its effectiveness and superiority over existing approaches.

**Weaknesses**
1. From the results in Table 1, the performance improvements brought by "w/ Aha" are relatively marginal, which reduces the solidness of the method proposed in this paper.
2. In Table 2, the method proposed in this paper shows poor performance in the Image Editing task, even falling behind many methods based on Stable Diffusion v1.5.
3. Since the GRPO algorithm has gained significant popularity, it may not be necessary for the authors to elaborate extensively on related content in the "Method" section.

---

> ### Author Rebuttal · Authors · 2025-07-30
>
> Thank you for your comprehensive comments. We are encouraged that our work is recognized to broaden the practical applications of MLLMs in the visual domain. We will explain your concerns as follows.
>
> > **(I) Weakness1:** The performance improvements brought by "w/ Aha" are relatively marginal.
>
> **A1:** Thank you for your question. We aim to clarify your concerns from two perspectives.
>
> **Firstly,** we want to claim that **activating the "Aha moment" yields clear capability gains**, especially in the widely-used benchmark GenEval. Specifically, as shown in Table 1, in GenEval, our model gains a 3-point boost in overall performance only considering the first-round text-to-image generation ("w/o Aha") compared to the backbone model Janus-Pro (from 0.80 to 0.83). When further unlocking Aha moments to introduce image regeneration ("with Aha"), the overall performance ascends an additional 3 points (from 0.83 to 0.86). Such improvement is not incremental on this benchmark.
>
>
> **Secondly**, although the performance gain introduced by "with Aha" over "w/o Aha"  is smaller than that of “w/o Aha” over the backbone model, it is expected and actually evidences the efficacy of our first-round image-generation capability. Specifically, during reinforcement learning, we encourage the model to satisfy the prompt **in as few rounds of image generation as possible**: **the fewer rounds it needs to produce a correct image, the larger the reward we provide.** For instance, if the model can generate a compliant image and perform an accurate self-evaluation in the first round, it could receive the maximum reward score.
>
> Consequently, compared with the backbone model, we have achieved a marked improvement in first-round-generated image quality. In the vast majority of cases, the model directly generates images that satisfy the requirements without any need for regeneration. Only when the initial generation is incorrect does the model invoke the Aha moment as the self-correction.  **Because such cases constitute only a small minority, the activation of the Aha moment acts as a refinement rather than a primary driver of performance.** Therefore, it is both reasonable and expected that the gain of "with Aha" over "w/o Aha" is smaller than that of "w/o Aha" over the backbone model.
>
> &nbsp;
>
> > **(II) Weakness2:** In Table 2, the method proposed in this paper shows poor performance in the Image Editing task, even falling behind many methods based on Stable Diffusion v1.5.
>
> **A2:** We apologize for not providing sufficient context about the benchmark, which may lead to some misunderstanding of the metrics.  In Table 2, PIE-Bench evaluates how well an image-editing model balances the trade-off between fidelity preservation and instruction following. Specifically, the first five metrics (Structure Distance and Background Preservation) in Table 2 evaluate fidelity preservation, i.e., how faithfully the model retains all details that should remain unchanged. While the final two metrics (CLIP Similarity) evaluate instruction following, that is, how accurately the model executes the requested edits.
>
> **Judging which model is superior requires considering both fidelity preservation and instruction following simultaneously**: all six metrics must be compared as an integrated whole, and focusing on a single dimension yields no meaningful insight. For example, though some baselines based on SD v1.5, such as MagicBrush, outperform our model on the first four metrics, they achieve this through an unacceptable shortcut: **in many editing cases, they simply return the unaltered source image as the post-edit output. This "zero-edit" trick grants them perfect fidelity scores,** yet it is clearly the opposite of what an editing model should do. Consequently, the last two metrics expose their poor editability and **render the high scores on the first four metrics meaningless.**
>
> Therefore, when evaluating a method's performance, we should assess whether it strikes a good balance between fidelity preservation and instruction following, rather than whether it achieves the highest scores on any single metric. Among baselines, **many methods based on SD v1.5, such as MagicBrush, obtain good fidelity preservation but the worst editability.** Only EditAR shows a relatively good trade-off between these two dimensions. In contrast, our **Janus-Pro-R1-Edit achieves better overall performance** and maintains a good balance between fidelity and editability, delivering the most effective performance.
>
> Moreover, as the community still lacks a standardized, objectively reliable automatic metric for image editing,  human judgment remains the de facto gold standard. To further showcase the superiority of our approach, we curated 100 diverse test cases, ranging from simple to highly complex edits. We conduct a rigorous **human evaluation**, where human evaluators rate each generated image on an overall 1-to-5 scale, jointly weighing diverse dimensions including fidelity preservation and instruction following.
>
> As shown in the following tables, **our model exhibits superior performance to open-source baseline methods for human evaluation of image editing** (only marginally underperforms GPT-4o), which demonstrates the strong performance of our model.
>
> **Table:** Human evaluation on image editing
> | Model | Human Score$\uparrow$ |
> |-|-|
> |InstructPix2Pix|2.7|
> |MagicBrush|2.5|
> |Ultraedit|3.1|
> |SeedX-Edit|2.8|
> |EditAR|3.0|
> |GPT-4o|**4.0**|
> |**Ours**|***3.5***|
>
> &nbsp;
>
> > **(III) Weakness3:** It may not be necessary for the authors to elaborate extensively on related content in the "Method" section.
>
> **A3:** Thank you for your suggestion. In fact, we have extended the application scope of GRPO, and the Method section primarily details how this algorithm is adapted to our specific task setting.
>
> Conventional GRPO is employed for single-modal token generation (generating text or visual tokens). In this paper, we extend it to interleaved text-image generation with dual-level rewards: comprehension rewards provide appropriate incentives for textual token generation, while generation rewards provide incentives for visual token generation. Besides, different samples within a group may entail varying numbers of image-generation and self-evaluation rounds. Consequently, applying GRPO in this setting introduces new challenges, such as how to integrate the two rewards and how to assign fair scores to samples with differing numbers of image-generation rounds.
>
> Thus, in the Method section, we provide a step-by-step elaboration of GRPO, **highlighting the precise modifications introduced to accommodate dual-level rewards and interleaved text–image generation**, enabling readers to understand our approach fully. In the revised manuscript, we will condense this section as you suggest.
>
> &nbsp;
>
> > **(IV) Question1:** How do the authors ensure the diversity of these prompts?
>
> **A4:** We totally constructed 20,0000 prompts for SFT. To maximize prompt diversity, we begin by hand-crafting hundreds of seed prompts and a modular set of instruction templates for prompt extension. The seed prompts with instruction templates span dozens of semantic axes (including counting, tone, spatial relationships, color, textual content, and many more), while also embracing a broad spectrum of descriptive styles. With Qwen as the tool LLM, we iteratively expand the seed prompts into a much larger and more heterogeneous prompt pool.
>
> Furthermore, to prevent the generated prompts from being uniformly short captions, we also instruct the LLM to upscale the short prompt into long and detailed descriptions. As a result, we construct ~200k prompts **spanning diverse content categories and descriptive styles, ranging from ultra-brief captions (<10 words) to richly detailed narratives (>150 words)**.
>
> &nbsp;
>
> > **(V) Question2:** Why is it necessary to use both Flux and Janus-Pro to generate images simultaneously, and what is the significance of image generation by Flux?
>
> **A5:** Janus-Pro and FLUX excel in different dimensions of text-to-image generation: FLUX delivers superior overall aesthetic quality and is more reliable on counting-related prompts, whereas Janus-Pro achieves stronger semantic alignment for prompts that specify spatial position or color. Meanwhile, the two models also produce different visual styles for the same prompt.
>
> By invoking both FLUX and Janus-Pro concurrently, **we leverage the complementary strengths of each model**. Across nearly every prompt, the final set of $M=18$ images contains **not only high-quality SFT samples but also deliberate variations in style and quality**. This simultaneously satisfies the diversity requirement and advances our goal of curating a quality-graded dataset.
>
> &nbsp;
>
> We hope we have addressed all of your concerns. Discussions are always open. Thank you!

---

### Official Review · Reviewer_yXuE · 2025-07-03

**Clarity:** 3
**Significance:** 3
**Originality:** 4
**Rating:** 4
**Confidence:** 4

**Summary:**

The paper proposes a post-training pipeline for unified MLLMs that turns text-to-image generation into a multi-turn image editing via CoTs , and shows Janus-Pro-R1, which is trained from a base model Janus-Pro. The paper shows that after a two-stage training pipeline (mixed SFT on three subtasks followed by GRPO-based RL), the model shows self-correction patterns (similar to the "Aha moment" in o1-like models) and is able to handle text-to-image and image editing better.

**Questions:**

1. Image editing may cause inconsistency, especially with many turns, where the mismatch and error will accumulate. Would this affect the final results, e.g., make longer CoT worse than short CoTs?

**Ethical Concerns:**

["NO or VERY MINOR ethics concerns only"]

**Limitations:**

Yes.

**Paper Formatting Concerns:**

N/A.

**Quality:**

3

**Strengths And Weaknesses:**

- Strengths

1. The motivation and idea is interesting. Turning text-to-image generation into a multi-round image editing bridged by (text-space) reasoning CoTs is reasonable.
2. The evaluation on public text-to-image generation benchmarks shows good empirical results. Janus-Pro-R1 tops or matches SOTA across three compositional T2I benchmarks and markedly improves PIE-Bench editing fidelity/editability trade-offs.
3. The authors provide good ablations on task synergy, reward models, and model size.

- Weaknesses

1. The RL optimises scores provided by InternVL2.5, which may suffer from reward hacking and can be inconsistent.
2. The prompt set is purely synthesized with Qwen, and the SFT images are generated by pre-trained diffusion models such as FLUX. This can cause worse diversity.
3. Turning text-to-image generation into multi-round editing task can introduce much higher computational overhead. The authors should discuss about the computational complexity in their paper.
4. Lack of user study on realism/editability, so practical impact is inferred solely from automated metrics.

---

> ### Author Rebuttal · Authors · 2025-07-30
>
> We sincerely thank you for your valuable comments. We are encouraged that our motivation and idea are recognized as interesting. We will explain your concerns as follows.
>
> > **(I) Weakness1:** The RL optimises scores provided by InternVL2.5, which may suffer from reward hacking and can be inconsistent.
>
> **A1:** Thank you for your question. Of course, we want the chosen RL reward score to mitigate reward hacking as much as possible and remain as aligned as possible with genuine human evaluation preferences. In selecting the reward model, we considered a wide range of candidates. After empirically comparing various vision-language models (VLMs), we found that InternVL2.5-26B is the best open-source VLM for most reliably assessing image–text semantic consistency while achieving the closest alignment with human evaluation.
>
> To illustrate the above conclusion, after collecting hundreds of images generated by Janus-Pro, we score them using human evaluation, and vision-language model (VLM) evaluation, including GPT-4o, InternVL2.5-8B/26B, LLaVA-1.6-34B, ShareGPT4V-13B, and Qwen2.5-VL-32B. We compare the score differences between the VLM evaluation results and the human evaluation results. As shown in the following table, **InternVL2.5-26B has a strong ability to judge fine-grained image-text consistency among open-source models** (even nearing the performance of GPT-4o), with its **evaluation results very close to those of human evaluation**. Thus, we chose InternVL2.5-26B as the evaluation model, effectively mitigating the issue of reward hacking.
>
> **Table:** Score difference between VLM evaluation and human evaluation.
> | VLM | Score Difference$\downarrow$ |
> |-|-|
> |GPT-4o| **0.06** |
> |InternVL2.5-26B| ***0.09*** |
> |InternVL2.5-8B | 0.20 |
> |LLaVA-1.6-34B | 0.17 |
> |ShareGPT4V-13B | 0.19 |
> |Qwen2.5-VL-32B | 0.11 |
>
> Of course, we are also considering using GPT-4o as the reward model, or combining different VLMs for different prompt configurations to provide reward scores, thereby further reducing reward hacking and improving evaluation consistency. However, this would inevitably increase training costs, so we defer this to future work.
>
> &nbsp;
>
> > **(II) Weakness2:**: The prompt set is purely synthesized with Qwen, and the SFT images are generated by pre-trained diffusion models such as FLUX. This can cause worse diversity.
>
> **A2:** **Firstly**, to maximize prompt diversity, we begin by hand-crafting hundreds of seed prompts and a modular set of instruction templates for prompt extension. The seed prompts with instruction templates span dozens of semantic axes (including counting, tone, spatial relationships, color, textual content, and many more), while also embracing a broad spectrum of descriptive styles. With Qwen as the tool LLM, we iteratively expand the seed prompts into a much larger and more heterogeneous prompt pool.
>
> Furthermore, to prevent the generated prompts from being uniformly short captions, we also instruct the LLM to upscale the short prompt into long and detailed descriptions. As a result, we construct ~200k prompts **spanning diverse content categories and descriptive styles, ranging from ultra-brief captions (<10 words) to richly detailed narratives (>150 words)**.
>
> &nbsp;
>
> **Secondly**, to generate the SFT images, we leverage both FLUX and Janus-Pro. Each model brings distinct strengths: FLUX excels in overall aesthetic quality and performs more reliably on counting-related prompts, whereas Janus-Pro achieves superior semantic alignment for prompts involving spatial position or color. Even within a single model, the multiple images generated for the same prompt can yield markedly different visual styles and semantic alignment.
>
> Therefore, for nearly every prompt, our final collection of $M=18$ images **not only includes high-quality SFT samples but also exhibits deliberate stylistic and quality divergence**, which simultaneously satisfies the diversity requirement and aligns with our goal of curating a quality-graded dataset.
>
> &nbsp;
>
> > **(III) Weakness3:**: The authors should discuss the computational complexity in their paper.
>
> **A3:** For computational overhead **during training**, we perform post-training (SFT+RL) on the pre-trained Janus-Pro model. The entire post-training lasted ten days: four days of SFT on 8 A100 GPUs and six days of RL on 32 A100 GPUs. **We consider this overhead well-justified**, as it equips Janus-Pro with the emergent ability to generate interleaved image–text sequences and, more importantly, unlocks its aha moment for visual generation.
>
> Regarding **inference**, we first acknowledge that activating the aha moment for image generation incurs higher time overhead: on average, *each round of image regeneration adds roughly 1.25× the cost of a standard text-to-image pass*. However, it is important to note that the model triggers a second round of image generation only if it self-evaluates that the initially generated image is semantically misaligned. For the vast majority of prompts, the model produces a faithful image in the first round, eliminating any need for regeneration. In these cases, the model only infers a handful of additional text tokens for self-evaluation, **incurring virtually no extra computational overhead compared to conventional text-to-image generation.**
>
> Image regeneration is invoked only when a semantic error is detected during self-evaluation. Although invoking image regeneration increases computational overhead, **it enables self-correction and yields output images that are more semantically aligned with the prompts** (see Table 1: "with Aha" vs. "w/o aha"). Echoing the design philosophy of models such as DeepSeek-R1 and GPT-4-o1, we view trading extra inference time for higher generation accuracy as a worthwhile investment.
>
> Furthermore, **the trade-off between image quality and computational overhead is also fully controllable** by the users, who can flexibly define a maximum number of regeneration rounds as needed. For instance, we limited the model to at most two regenerations (a maximum of 3-round image generations) to deliver near-optimal visual quality while keeping time overhead within an acceptable range.
>
> &nbsp;
>
> > **(IV) Weakness4**: Lack of user study on realism/editability, so practical impact is inferred solely from automated metrics.
>
> **A4:** We sincerely apologize that practical impact is inferred solely from automated metrics. In our paper, besides automated evaluation, we provide qualitative comparisons of text-to-image generation in Appendix Fig.1 and image editing in MainPDF Fig.4. To further provide a solid user study on realism/editability, we construct a comprehensive test set, including 100 image editing test cases and 100 text-to-image generation test cases, covering a wide range of editing instructions and prompt categories.
>
> We conduct a rigorous human evaluation, where human evaluators rate each generated image on an overall 1-to-5 scale, jointly weighing diverse dimensions including realism and editability. As shown in the following tables, **for human evaluation, our model exhibits superior performance to open-source baseline methods in both tasks** (only marginally underperforms GPT-4o), which demonstrates the strong performance of our model.
>
> **Table:** Human evaluation on text-to-image generation
> | Model | Human Score$\uparrow$ |
> |-|-|
> |FLUX.1-dev|3.7|
> |Sana-1.5|3.3|
> |Show-o|2.6|
> |Janus-Pro|3.0|
> |T2I-R1|3.4|
> |GPT-4o|**4.2**|
> |**Ours**|***4.0***|
>
> **Table:** Human evaluation on image editing
> | Model | Human Score$\uparrow$ |
> |-|-|
> |InstructPix2Pix|2.7|
> |MagicBrush|2.5|
> |Ultraedit|3.1|
> |SeedX-Edit|2.8|
> |EditAR|3.0|
> |GPT-4o|**4.0**|
> |**Ours**|***3.5***|
>
> &nbsp;
>
> > **(V) Question1**: Image editing may cause inconsistency, especially with many turns, where the mismatch and error will accumulate. Would this affect the final results, e.g., make longer CoT worse than short CoTs?
>
> **A5:** Thank you for the question; we would like to clarify potential misunderstandings from two perspectives.
>
> **(1)** For text-to-image, the generated image only needs to align with the semantics of the user prompt. After reformulating text-to-image generation as a multi-round image-editing process, we regenerate the image in each round solely to rectify any semantic misalignment between the current image and the prompt. **Inconsistencies in prompt-unrelated regions across the images from different rounds do not create mismatches with the prompt; consequently, accumulated inconsistency does not degrade the performance of text-to-image generation.**
>
> The table below reports the results on Geneval and T2I-CompBench when the aha moment is triggered from 0 to 2 times. It can be seen that **the performance of longer CoTs (triggering more Aha moments) still surpasses that of shorter CoTs.**
>
> **Table:** Performance on text-to-image generation with longer/shorter CoTs.
>
> | CoT Length | GenEval (Overall) | T2ICompbench (Avg) | DPG-Bench |
> |-|-|-|-|
> |w/o Aha|0.83|70.3|85.02|
> |with Aha (1time at most)|0.85|72.0|85.41|
> |with Aha (2times at most)|**0.86**|**72.7**|**85.57**|
>
> &nbsp;
>
> **(2)** After upgrading the t2i model to multi-turn image editing, we observe that the model also becomes better aligned with the image-editing task itself. While image editing additionally necessitates preserving image fidelity, when leveraging Janus-Pro-R1 for this task, we do NOT introduce Aha moments with image regeneration for each turn of editing to avoid accumulating inconsistencies. As shown in Fig.4, compared with baseline methods, for each turn of editing, the output images of our model consistently resemble the source images while remaining coherent with the instructions.
>
> &nbsp;
>
> Thank you once again for your constructive suggestion! We are grateful for the opportunity to improve our work based on your insightful comments.

---

> > ### Comment · Reviewer_yXuE · 2025-08-09
> >
> > Thank the authors for their response. The additional human evaluations and ablations are essential for understanding and evaluating the proposed method, so I would strongly recommend the authors to include them in their final paper. Moreover, I would still suggest the authors to analyze the complexity and compare with non-reasoning image generation in their final paper. I will keep my positive score.

---

> > > ### Author Response · Authors · 2025-08-09
> > >
> > > We are grateful for your suggestions, which have significantly contributed to improving our paper! We will integrate the refined analyses and the new experimental results into the next version of our manuscript.

---

### Author Response · Authors · 2025-08-05
**General Response**

Dear Reviewers, ACs, and SACs:

We sincerely thank you for the precious time and insightful feedback, which has significantly strengthened our manuscript! Overall, we are encouraged that you find that:

- The motivation and idea are *interesting and reasonable*, with the evaluation showing good empirical results supported by comprehensive ablation studies. (Reviewer ``yXuE``)

- The method enables a more *intelligent and coherent* image generation process, *broadening the practical applications of MLLMs* in the visual domain, with SOTA performance demonstrating its *effectiveness and superiority* over existing approaches. (Reviewer ``xYFR``)

- The method is *technically sound with clear writing*. (Reviewer ``Fzpc``)

- The work provides a *working implementation* for training a unified model, analyzing the synergy between visual comprehension and generation thoroughly. It also presents a *compelling example* of unifying these two capabilities, and the paper’s *clear presentation* further helps gauge the contribution within the current scientific progress. (Reviewer ``oSZo``)

&nbsp;

To address the concerns raised by the reviewers, overall we have conducted several additional experiments:
- Empirically explaining why we chose InternVL2.5-26B as our reward model and demonstrating that our reward calculation is reasonable.
- Introducing human evaluation for text-to-image generation and image editing to corroborate the effectiveness of our method.
- Conducting more experiments on visual comprehension tasks to show the synergetic gains across visual comprehension and generation.
- Conducting step-by-step evaluation in text-to-image generation to reveal consistent performance improvements throughout the CoT process.
- Comparing with more previous composition and iterative image generation methods, further confirming our superior performance.

We have also clarified the following key points:
- Detailing the strategies to ensure diversity in our SFT data.
- Analyzing the computational complexity of our method.
- Providing an in-depth explanation of the experimental results reported in Tables 1 and 2 to show the superior performance of our method.
- Presenting a theoretical discussion of the advantages of unifying visual comprehension and generation within a single model over using separate models.

&nbsp;

These experiments and clarifications will be integrated into the main body or the appendix of our paper. Once again, we sincerely thank all reviewers for the valuable suggestions! With three days remaining in the Author-Reviewer phase, we warmly welcome any further questions or suggestions. Discussions are always open! Thank you!

&nbsp;

*Best regards,*

*NeurIPS 2025 Conference Submission13390 Authors*

---

### Note · Authors · 2025-08-16

We extend our heartfelt gratitude to the reviewers for their insightful feedback and constructive suggestions.

Our work is highly praised by reviewers for its **interesting and reasonable** motivation and idea, as well as its **technically sound** implementation with **clear writing**. Reviewers highlight that Janus-Pro-R1 enables a more **intelligent and coherent image generation process**, significantly **broadening the practical applications of MLLMs** in the visual domain. Reviewers also note that our work provides a **working implementation** for training a unified model, thoroughly analyzing the synergy between visual comprehension and generation. Furthermore, they acknowledge that our model demonstrates **state-of-the-art performance**, showing its effectiveness and superiority through **comprehensive evaluation and ablation studies**.

&nbsp;

During the discussion phase, we addressed all raised concerns with concrete clarifications and additional experiments:

- **Training Data Diversity:** We provide a detailed explanation of how we curate the training data to guarantee its rich diversity.

- **Superior Performance:** We first provide an in-depth explanation of the results reported in Tables 1 and 2, demonstrating that our model outperforms all other baselines. We also extend the comparison to more baselines and further conduct a step-by-step evaluation, confirming our superior performance.

- **Reword Model Designing:** We empirically explain why we chose InternVL2.5-26B as our reward model, demonstrating that our reward calculation is reasonable.

- **Advantages of Unifying Visual Comprehension and Generation in a Single Model**: We conduct more experiments on visual comprehension tasks to show the synergetic gains across visual comprehension and generation. We also give a theoretical discussion of the advantages of unifying visual comprehension and generation within a single model over using separate models.

- **Human Evaluation**: We introduce human evaluation for both text-to-image and image editing tasks to corroborate the effectiveness of our method.

These experiments and clarifications will be incorporated into the next version of our paper. We believe this work delivers a working implementation of collaborating visual comprehension and generation within a unified model, and we hope our findings will inspire further research. Once again, we sincerely thank all reviewers for the precious time and insightful feedback!

---

### Decision · Program_Chairs · 2025-09-17

**Decision:**

Accept (poster)

**Comment:**

The reviewers generally found the paper to be well motivated, clearly written, and technically sound, with a novel framing of text-to-image generation as a multi-round editing process guided by reasoning chains. They highlighted the strong empirical performance across multiple benchmarks, including state-of-the-art or competitive results on compositional T2I tasks and improved trade-offs in image editing. Reviewers also appreciated the thorough ablations on model components and training setup, as well as the broader potential of the method to unify image generation and editing within multimodal LLMs. While some reviewers noted concerns about reward model reliance, synthesized training data, computational overhead, and relatively modest or uneven improvements in some evaluations, the overall consensus leaned positive, with one strong accept and multiple borderline accepts.

During the rebuttal, the authors provided additional ablations and human evaluations that helped address key concerns around the reliability of results and practical impact. Clarifications regarding judge model choices, performance on harder editing cases, and the role of task synergy strengthened confidence in the method’s soundness and generality. Given the overall novelty, solid empirical results, and clarified responses, the final recommendation is accept.